# Mutant Huntingtin stalls ribosomes and represses protein synthesis in a cellular model of Huntington disease

Mehdi Eshraghi [1], Pabalu P. Karunadharma[2], Juliana Blin [3], Neelam Shahani [1], Emiliano P. Ricci [3], Audrey Michel[4], Nicolai T. Urban [5], Nicole Galli[1], Manish Sharma[1], Uri Nimrod Ramírez-Jarquín[1], Katie Florescu[1], Jennifer Hernandez[1] & Srinivasa Subramaniam[1✉]

The polyglutamine expansion of huntingtin (mHTT) causes Huntington disease (HD) and neurodegeneration, but the mechanisms remain unclear. Here, we found that mHtt promotes ribosome stalling and suppresses protein synthesis in mouse HD striatal neuronal cells. Depletion of mHtt enhances protein synthesis and increases the speed of ribosomal translocation, while mHtt directly inhibits protein synthesis in vitro. Fmrp, a known regulator of ribosome stalling, is upregulated in HD, but its depletion has no discernible effect on protein synthesis or ribosome stalling in HD cells. We found interactions of ribosomal proteins and translating ribosomes with mHtt. High-resolution global ribosome footprint profiling (Ribo-Seq) and mRNA-Seq indicates a widespread shift in ribosome occupancy toward the 5′ and 3′ end and unique single-codon pauses on selected mRNA targets in HD cells, compared to controls. Thus, mHtt impedes ribosomal translocation during translation elongation, a mechanistic defect that can be exploited for HD therapeutics.

[1] The Scripps Research Institute, Department of Neuroscience, Jupiter, FL, USA. [2] The Scripps Research Institute, Genomic Core, Jupiter, FL, USA. [3] Laboratory of Biology and Cellular Modelling at Ecole Normale Supérieure of Lyon, RNA Metabolism in Immunity and Infection Lab, LBMC, Lyon, France. [4] RiboMaps Ltd, Cork, Ireland. [5] The Max Planck Neuroscience Institute, Jupiter, FL, USA. ✉email: ssubrama@scripps.edu

Ribosomes move one codon at a time along mRNA during protein synthesis, a fundamental process in all living cells. During this translocation (aka ribosome movement), the ribosomes pause for a variety of reasons, such as codon usage, peptide properties, mRNA structure, and tRNA availability, or solely due to a continuously changing cellular demand[1–7]. This pause can be a transient event (e.g., during the insertion of the signal peptide into the endoplasmic reticulum), or it can be permanent (e.g., during a wrong codon–anticodon pairing)[8]. Regardless, the mechanisms of ribosome pausing (aka ribosome stalling) or its dysregulation in neurodegenerative disease remain poorly understood. Elucidation of the mechanisms governing ribosome stalling should provide an opportunity to develop effective therapeutic interventions.

Ribosome stalling in neurodegenerative disease is associated with several different proteins. For example, GTP-binding protein 2 (GTPBP2), identified after an N-ethyl-N-nitrosourea mutagenesis screen, is known to regulate stalling at the AGA codon, and GTPBP2 deletion promotes a spontaneous ataxia-like neurodegenerative phenotype[9]. Similarly, TAR DNA-binding protein 43 (TDP-43), which promotes amyotrophic lateral sclerosis (ALS) and frontotemporal lobar degeneration (FTLD), interacts with stalled ribosomes during cellular stress and inhibits protein synthesis[10,11]. Mutations in other mRNA-binding proteins (RBPs), such as SMN, FUS, and Ataxin-2 (proteins that also regulate protein synthesis but with unclear mechanisms), can also promote neurodegenerative diseases[12–15]. The Fmrp, another RBP, can bind to ribosomes and regulate ribosome stalling[16–18]. The depletion of *Fmr1*, as observed in Fragile X syndrome, results in increased protein synthesis that is linked to synaptic and behavioral defects and that can also elicit the neurodegenerative phenotype in Fragile X-associated tremor/ataxia syndrome[19,20]. Deficits in rescuing ribosome stalling may also promote neurodegeneration[2,21]. Major unanswered questions include how ribosome stalling is directly regulated by physiological signals and how its dysregulation affects neurodegenerative disease processes.

Huntingtin (HTT), a ubiquitously expressed protein found throughout the nervous system and in non-neural tissues. Mice with deletion of the Htt gene, *Hdh*, die around embryonic day 8.5–10.5 before the full emergence of the nervous system, indicating a role for this gene in cell survival and neurogenesis[22–25]. Conditional deletion of Htt in the mouse brain results in a defect in corticostriatal development, as well as induces hyperactivity, acute pancreatitis, and age-dependent neurodegenerative-like phenotype[26–28]. These results indicate that normal HTT plays essential role in the organism's developmental and adult brain functions.

The CAG expansion of HTT gene results in mutant HTT (mHTT) protein, which causes Huntington's disease (HD), a debilitative autosomal-dominant brain disorder with worldwide distribution[29–31]. Although mHTT does not appear to perturb development in mouse models, it has been shown to interfere with cortical neurogenesis in the human fetal brain[32,33].

In humans, HD onset and severity of symptoms depend on the number of CAG repeats in *HTT*. Brain pathology and magnetic resonance imaging studies show early severe damage to the striatum[34–36]. As the disease progresses, the damage extends to the cortex and multiple CNS and peripheral regions, leading to motor dysfunctions, weight loss, and energy deficits[37–40]. The majority of HD patients are heterozygous, but some homozygous patients experience a severe clinical course, such as rapid striatal atrophy and decline in motor, cognitive, and behavioral skills[41,42]. These HD deficits can emanate from one or more effects of mHTT and its proteolytically cleaved fragments on several functions, such as vesicle- and microtubule-associated protein/organelle transport, transcription, autophagy, and sphingosine and cysteine

metabolism, oxidative stress, calcium signaling, as well as effects on tissue maintenance, secretory and inflammatory pathways, and cell division[43–69]. Altered ribosomal functions and association of HTT and mHTT with translating ribosomes were reported in HD model systems and HD patient-tissue[51,70–78]. But the evidence for the role(s) or the mechanism(s) of HTT in the regulation of protein synthesis is limited. Here, we report that mHtt suppresses protein synthesis via mechanisms involving ribosome stalling potentially occurring during elongation.

## Results

**Ribosome stalling and suppression of protein synthesis in HD cells.** Protein synthesis is regulated in a cell-type-specific manner; therefore, we employed homogenous HD knock-in cell models for our translation studies[79,80]. We investigated mRNA translation using genetically precise striatal neuronal cells that express a targeted insertion of a chimeric human–mouse exon 1 with 7/7 CAG (ST*Hdh*$^{Q7/Q7}$, control), 7/111 CAG (ST*Hdh*$^{Q7/Q111}$, HD-het), and 111/111 CAG (ST*Hdh*$^{Q111/Q111}$, HD-homo) repeats[80]. We used polysome profiling to estimate mRNA loading onto translating polyribosomes. We identified a high polysome/monosome (PS/MS) ratio in the HD-homo compared to the HD-het or control cells (Fig. 1A, B) by integrating the area under the curve from raw profiles (Supplementary Fig. S1). We hypothesized that the high PS/MS ratio in HD-homo cells reflected a more actively translating mRNA in the HD-homo cells. We examined this possibility by measuring protein synthesis with SUnSET, a nonradioactive puromycin/antibody-based tool[81]. However, we found diminished puromycin incorporation, reflective of reduced mRNA translation (protein synthesis), in the HD-homo (~40%) and HD-het cells (~20%) compared to the control cells (Fig. 1C, D). As with puromycin, we found diminished incorporation of radiolabeled [$^{35}$S]-methionine into newly synthesized proteins in the HD-homo cells (~40%) compared to the control cells (Supplementary Fig. S2). Human HD-het fibroblasts also showed a significant reduction in protein synthesis compared to unaffected controls (Supplementary Fig. S3).

We then hypothesized that the high PS/MS ratio in HD-homo striatal cells, despite the diminished protein synthesis, could reflect a pause in ribosome movements, which would lead to a more diminished rate of translation elongation in the HD-homo than in the control cells. We tested this hypothesis in ribosome run-off experiments with harringtonine, a compound that inhibits the initiation of mRNA translation without affecting the ribosomes that have cleared the start codon[82–85]. Analysis of the ribosome profiling of harringtonine-treated cells, therefore, allows a determination of whether the cells are experiencing ribosome stalling. If ribosomes are stalled, then the profile will show a lower MS peak in cells that are stalled (i.e., cells that have a high PS/MS ratio) than in cells in which ribosomes are not stalled[86]. If ribosomes run faster, then the profile will show an increased MS peak (i.e., cells that have a low PS/MS ratio) than in cells in which ribosomes are moving slower.

We found that harringtonine treatment (~2 min) resulted in a rapid increase in the 80S (MS) peak in control striatal cells (Fig. 1E, arrow) compared to the HD-homo striatal cells. At this point, the polysome profile in the HD-homo cells remained high compared to the control cell profile (Fig. 1E, inset e2; arrowhead). These data indicated that ribosomes ran slower in HD cells than in control cells. After 5 min of harringtonine treatment, the polysome profile still appeared higher in HD cells than in the control cells (Fig. 1E, inset e3, arrowhead). By 8 min, the ribosomes appeared to have completed their translation (run-off) similarly in both the HD-homo and the control cells (Fig. 1E, inset e4, arrowhead). The PS/MS ratio between control and HD

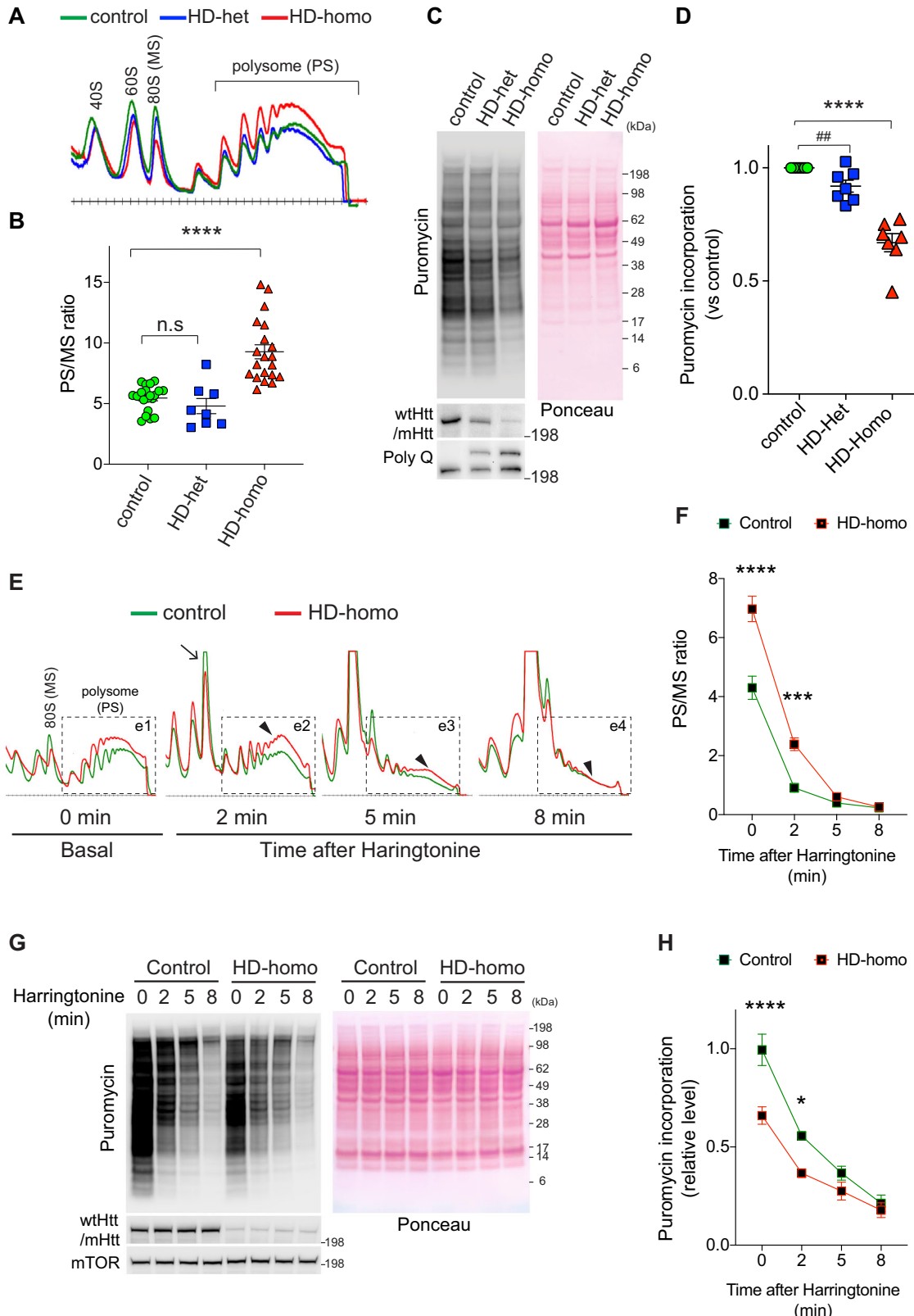

cells showed a significant correlation after the harringtonine treatment (Fig. 1F). Moreover, despite their high PS/MS ratio (Fig. 1E, F), the HD cells showed diminished protein synthesis under the harringtonine treatment (Fig. 1G, H), indicative of stalled and slowly elongating ribosomes in HD.

We also found a much slower ribosome runoff in HD cells than in controls in the presence of puromycin (Supplementary Fig. S4A, B), another compound that can be employed to assess the ribosome runoff[16,87,88]. Thus, the harringtonine and puromycin experiments both showed that polysome depletion from mRNA occurs much more slowly in HD cells than in control cells. Collectively, these data indicate that the diminished protein synthesis in HD striatal cells potentially occurs due to an inhibition of translational elongation caused by ribosome stalling.

**Fig. 1 Suppression of protein synthesis and ribosome stalling in HD cells. A** Representative polysome profile of control, HD-het and HD-homo striatal cells obtained by using sucrose density gradient fractionation. **B** Quantification of polysome to monosome (PS/MS) ratio in polysome profiles from **A** (area under the curve). Data are mean ± SEM (control, $n = 20$; HD-Het, $n = 8$; HD-Homo, $n = 20$ independent experiments) ****$P < 0.0001$ by one-way ANOVA followed by Tukey's multiple comparison test. **C** Representative immunoblots of metabolic labeling of protein synthesis, using puromycin, and its quantification (**D**) in mouse striatal cells. Ponceau staining of the blots was used to quantify the total protein signal in each lane. Data are mean ± SEM ($n = 7$, independent experiments) ##$P < 0.01$ by two-tailed Student's $t$ test and ****$P < 0.0001$ by one-way ANOVA followed by Tukey's multiple comparison test. **E** Representative polysome profiles obtained from control and HD-homo striatal cells at basal (0 min) and after ribosome run-off assay with harringtonine (2 µg/ml, 2, 5, and 8 min). Area e1–e4 shows ribosome movement between control (green) and HD-homo (red) cells. **F** Quantification of polysome to monosome (PS/MS) ratio in polysome profiles from **E** (area under the curve). Data are mean ± SEM (at 0 min, $n = 8$; 2 min, $n = 8$; 3 min, $n = 4$; 8 min, $n = 6$ independent experiments), ***$P < 0.001$ and ****$P < 0.0001$, by two-way repeated measures ANOVA, Bonferroni's multiple comparisons test. **G** Representative immunoblots of metabolic labeling of protein synthesis, using puromycin, in control and HD-homo striatal cells at basal (0 min) and 2, 5, and 8 min after harringtonine treatment and quantification (**H**). Ponceau staining of the blots was used to quantify the total protein signal in each lane. Data are mean ± SEM ($n = 7$, independent experiments), *$P < 0.05$ and ****$P < 0.0001$ by two-way repeated measures ANOVA, Bonferroni's multiple comparisons test. Exact $P$ values are reported in the Source Data file. Source data are provided as a Source Data file.

**Depletion of Htt enhances protein synthesis and increases the speed of ribosome translocation**. We then hypothesized that if mHtt is directly responsible for inhibiting protein synthesis in striatal neuronal cells, as observed in Fig. 1, then its depletion should increase protein synthesis. We tested this hypothesis using CRISPR/Cas9 tools to deplete wtHtt in control cells and mHtt in HD-homo cells. We found depletions of ~60% wtHtt and ~80% mHtt in these cells (Fig. 2A, B). Consistent with Fig. 1C, G, protein synthesis was lower in the CRISPR ctrl HD cells than in the CRISPR ctrl cells; however, depletion of either wtHtt or mHtt resulted in a significant increase in protein synthesis, as measured by puromycin incorporation (Fig. 2A, C). The fold change of the increase in protein synthesis was similar between the wtHtt or mHtt-depleted cells (Fig. 2C). These data indicate that both wtHtt and mHtt inhibit protein synthesis.

We then hypothesized that the enhancement of protein synthesis by depletion of Htt was accompanied by increased ribosomal movements. We evaluated this possibility by conducting ribosome run-off assays in wtHtt-depleted and mHtt-depleted cells under basal (vehicle) and harringtonine-treated conditions. Under the basal conditions, we found no apparent differences in the ribosome profiles, nor did we note any changes in the PS/MS ratio between the CRISPR ctrl and CRISPR wtHtt-depleted control cells (Fig. 2D, E). By contrast, in the presence of harringtonine, the CRISPR wtHtt-depleted cells showed a more rapid increase in MS peak compared to the wtHtt-intact control cells, as well as a significant decrease in the PS/MS ratio (Fig. 2D, F, arrow). Similarly, under basal conditions, the PS/MS ratio was unaltered in CRISPR ctrl or CRISPR mHtt-depleted HD-homo cells (Fig. 2G, H), but the presence of harringtonine caused a significant decrease in the PS/MS ratio in the CRISPR mHtt-depleted HD-homo cells compared to mHtt-intact HD-homo cells (Fig. 2G, I). These data indicate that ribosomes run faster when Htt is depleted.

We next compared the effects of harringtonine treatment on polysome profiles in CRISPR wtHtt versus CRISPR mHtt-depleted cells (Fig. 2J). We found a higher PS/MS ratio in harringtonine-treated CRISPR mHtt-depleted cells compared to CRISPR wtHtt-depleted cells (Fig. 2J, K). These data further support the notion that ribosomes run much more slowly in HD cells. Collectively, these data reveal that mHtt inhibits protein synthesis most likely at the level of elongation by promoting ribosome stalling.

**Htt directly inhibits protein synthesis in vitro**. We investigated whether Htt can directly modulate protein synthesis. We used recombinant human full-length (FL)-wtHTT (23Q) and FL-mHTT (48Q), which were purified from HEK293 cells and appeared >95% pure on Coomassie gels (Fig. 3A). We tested the

effect of these HTT proteins in a rabbit reticulocyte-based in vitro translation (IVT) assay system that measured luciferase synthesis and activity as the relative luciferase unit (RLU). At a 50 nM concentration, both the FL-wtHTT (23Q) and FL-mHTT (48Q) proteins caused a ~40% reduction in the RLU, compared to a BSA control (Fig. 3B). At 200 nM concentration, we found ~95% reduction in the RLU but a significantly stronger inhibition by FL-mHTT (48Q) than by FL-wtHTT 23Q (Fig. 3B). This stronger inhibition at a higher concentration led us to look for a dose-dependent effect of HTT proteins on protein synthesis. We found that both FL-wtHTT (23Q) and FL-mHTT (48Q) robustly blocked the luciferase expression (RLU) in a concentration-dependent manner (Fig. 3C). Notably, mHTT failed to inhibit luciferase activity if it was added for 5 min after the IVT reaction was completed (90 min), suggesting that mHTT actively engages the protein synthesis machinery to inhibit luciferase synthesis. We then tracked luciferase activity at different timepoints during the IVT reaction and found a reduced relative luminescence signal. This reduction was much stronger in the presence of the FL-mHTT (48Q) protein than in the presence of FL-wtHTT (23Q) protein at timepoints of 45 and 75 min during the IVT assay (Fig. 3D). Collectively, these data consistent with a previous report[75], indicate that both wtHTT and mHTT can directly inhibit protein synthesis, although the inhibitory effect may be slightly stronger for mHTT than for wtHTT.

**Fmrp, a known inhibitor of stalling, is upregulated in HD**. Given the evidence that ribosomes were apparently stalled in HD cells, we wanted to identify the nature of the ribosome-bound translating mRNAs, as this knowledge could help in identifying the mRNA transcripts that are stalled as well as in understanding the mechanisms of ribosome stalling in HD. We addressed this by isolating mRNAs from the slowly translating polysomes in HD-homo cells and comparing them to mRNAs from control cells using the harringtonine-based ribosome run-off assay (RRA), followed by mRNA-Seq (PS-RRA-mRNA-Seq) (Fig. 4A). This approach identified ~1157 targets (Fig. 4B and Supplementary Data 1) that showed high mRNA abundance (red dots) and ~1248 targets that showed low mRNA abundance (blue dots) in the polysome of HD-homo cells ($P < 0.05$) compared to the control polysome. Of these targets, we noticed higher levels of Fmrp (encoded by *Fmr1*), a known inducer of ribosome stalling[18], in the PS-RRA-RNA-Seq from the HD-homo cells than from the control cells ($P$adj = 0.01), whereas the levels of *Gapdh*, *mTOR*, *Eif2α*, and *Rps27* were not significantly altered (Fig. 4C, upper panel). The qPCR analysis of the ribosome fractions (MS and PS) not treated with harringtonine showed similar levels of *Fmr1*, *Gapdh*, and *Rps27* mRNA in both the control and

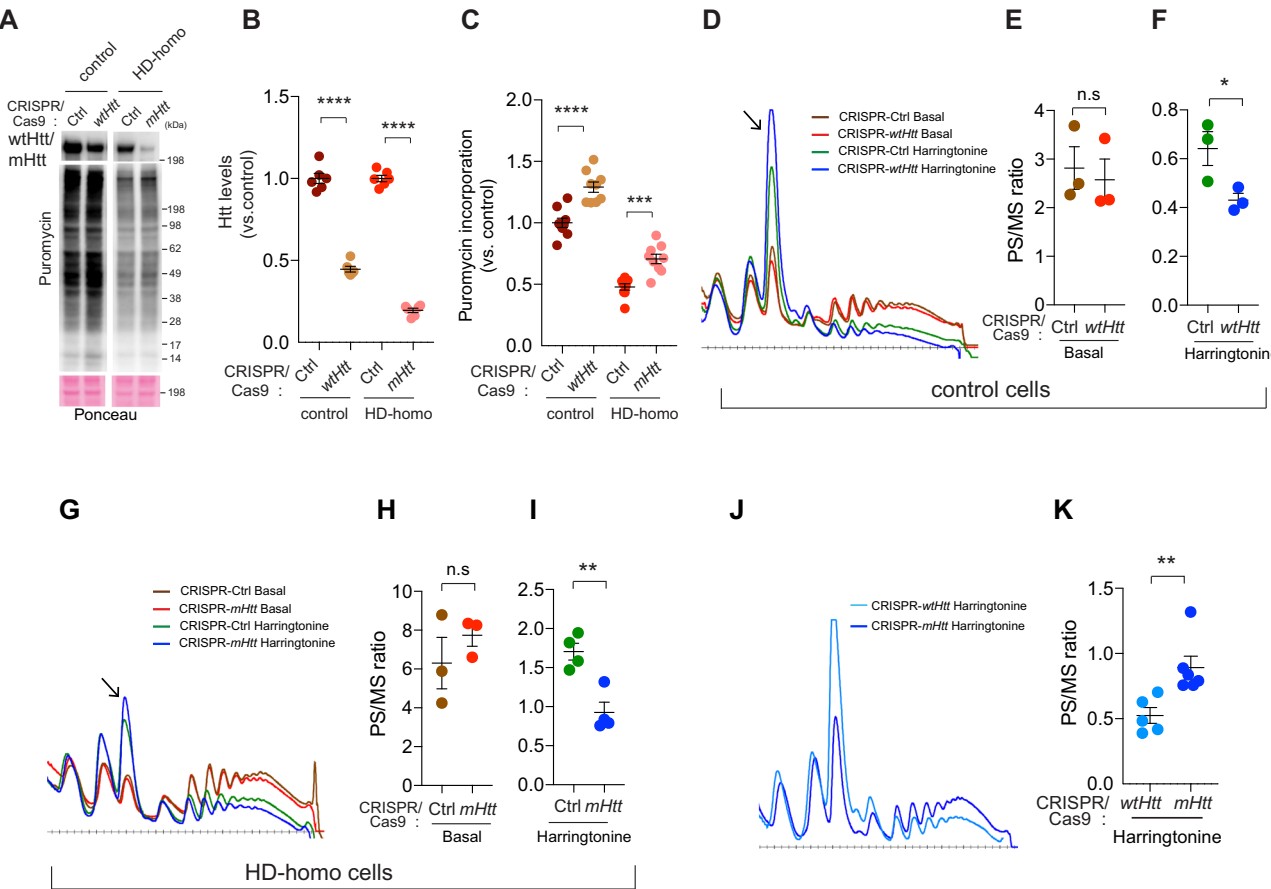

**Fig. 2 Depletion of mHtt enhances protein synthesis and increases the speed of ribosome translocation. A** Representative immunoblots performed on CRISPR/Cas9 *Htt*-depleted mouse striatal cells after puromycin metabolic labeling. **B**, **C** quantifications of the blots from **A** showing levels of Htt (**B**), puromycin incorporation (**C**) in control (Ctrl) and *Htt*-depleted control, HD-homo striatal cells. Data are mean ± SEM ($n = 9$ independent experiments), ***$P < 0.001$ and ****$P < 0.0001$ one-way ANOVA followed by Bonferroni's multiple comparisons test. **D**–**F** Representative polysome profiles (**D**) obtained from *wtHtt*-depleted mouse striatal cells before (basal, **E**) and after ribosome run-off assay using harringtonine (2 µg/ml, 2 min, **F**) and their corresponding quantification of polysome to monosome (PS/MS) ratios (area under the curve). Data are mean ± SEM ($n = 3$ independent experiments), *$P < 0.05$ by two-tailed Student's *t* test, n.s not significant. **G**–**I** Representative polysome profiles (**G**) obtained from *mHtt*-depleted mouse striatal cells before (basal, **H**) and after ribosome run-off assay using harringtonine (2 min, **I**) and their corresponding quantification of polysome to monosome (PS/MS) ratios (area under the curve). Data are mean ± SEM (**H**, $n = 3$; **I**, $n = 4$ independent experiments), **$P < 0.01$ by two-tailed Student's *t* test, n.s not significant. **J**, **K** Representative polysome profiles (**J**) obtained from *wtHtt*- and *mHtt*-depleted mouse striatal cells after ribosome run-off assay using harringtonine (2 min), and their corresponding quantification of polysome to monosome (PS/MS) ratios (**K**). Data are mean ± SEM (*wtHtt* depleted, $n = 5$; *mHtt* depleted, $n = 6$ independent experiments), **$P < 0.01$ by two-tailed Student's *t* test. n.s not significant. Exact $P$ values are reported in the Source Data file. Source data are provided as a Source Data file.

HD-homo cells. These results indicate that *Fmr1* mRNA strongly binds to ribosomes in HD (Fig. 4C, lower panel).

Because *Fmr1* mRNA is associated with polysomes in HD cells, we hypothesized that *Fmr1* mRNA is stalled in HD and that its protein levels should be low. However, contrary to this assumption, we found that Fmrp protein levels were significantly upregulated in the HD-homo cells, as well as in human HD patient striatum, whereas the total *Fmr1* mRNA was differentially altered (Fig. 4D, E). Collectively, PS-RRA-mRNA-Seq data suggested that Fmrp is upregulated in HD.

**Fmrp does not affect mHtt-mediated protein synthesis or ribosome stalling in HD cells.** As Fmrp is upregulated in HD and is a known inducer of ribosome stalling, we hypothesized that Fmrp is likely involved in ribosome stalling in HD cells. We examined the role of Fmrp using CRISPR-*Fmr1* to deplete *Fmr1* in control and HD-homo cells; this resulted in ~80% reduction in the Fmrp protein levels (Fig. 5A, B). We found that Fmrp depletion in control cells increased protein synthesis, as measured

by enhanced puromycin incorporation, whereas, surprisingly, it had no discernable effect on protein synthesis in the HD-homo cells (Fig. 5A, C).

We further dissected the Fmrp role in HD cells by generating a double knockout of mHtt/Fmrp, using CRISPR/Cas9. We found that cells in which mHtt alone were depleted showed a significantly increased protein synthesis, whereas cells in which Fmrp alone was depleted or in which both mHtt and Fmrp were depleted showed protein synthesis levels similar to those of cells in which mHtt alone was depleted (Fig. 5D, E). We then used harringtonine to examine whether Fmrp depletion had any effect on ribosome stalling in HD cells. The mHtt depletion enhanced the ribosome runoff (Fig. 5F, G), as before (Fig. 2G, I). However, Fmrp depletion did not significantly alter the ribosome runoff (Fig. 5F, G). These data indicated that (i) Fmrp upregulation might not impact overall protein synthesis inhibition in the cultured HD cells, and (ii) mHtt acts upstream or independently of Fmrp to promote ribosome stalling and inhibit protein synthesis in HD cells.

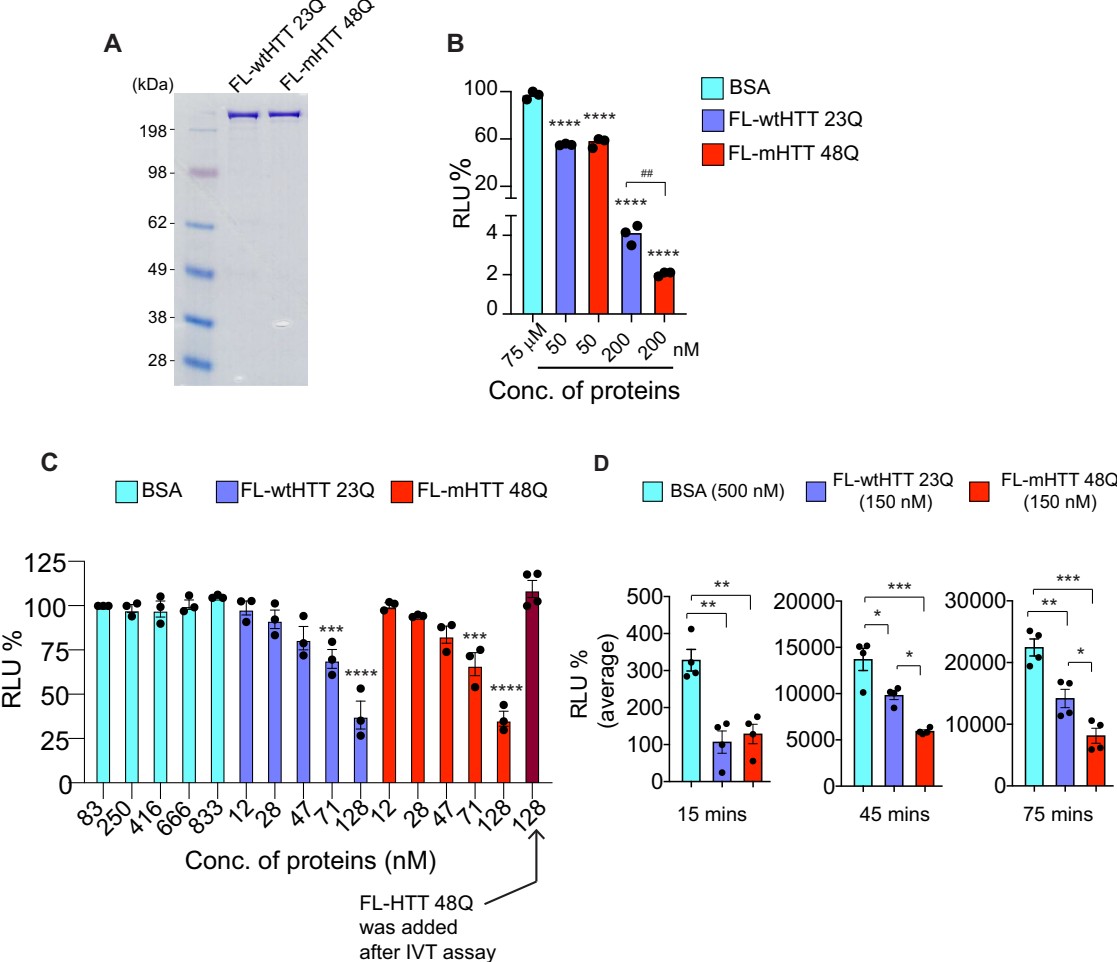

**Fig. 3 mHTT inhibits protein synthesis in vitro. A** Coomassie gel showing FL-wtHTT 23Q (250 ng) and FL-mHTT 48Q (250 ng), $n = 3$ independent experiments. **B** Shows the percentage of relative luciferase unit (RLU) in presence of an indicated concentration of recombinant FL-wtHTT and FL-mHTT. HTT proteins normalized to the maximum luminescence in BSA (75 μM). Data are mean ± SEM ($n = 3$ independent experiments). ****$P < 0.0001$ by one-way ANOVA followed by Tukey's multiple comparison test. ##$P < 0.01$ by two-tailed Student's $t$ test. **C** In vitro translation (IVT) assay in presence of different concentrations of recombinant FL-wtHTT 23Q and FL-mHTT 48Q or BSA. Bar graphs depict the percentage of relative luciferase unit (RLU) normalized to the maximum luminescence in BSA (83 nM). Data are mean ± SEM ($n = 3$ independent experiments). ***$P < 0.001$, ****$P < 0.0001$ by one-way ANOVA followed by Bonferroni's multiple comparisons test. **D** Shows luciferase activity at 15, 45, and 75 min of IVT reaction with 150 nM recombinant FL-wtHTT 23Q and FL-mHTT 48Q proteins or BSA (500 nM). Data are mean ± SEM ($n = 4$ independent experiments). *$P < 0.05$, **$P < 0.01$, ***$P < 0.001$, by one-way ANOVA followed by Tukey's multiple comparison test. Exact $P$ values are reported in the Source Data file. Source data are provided as a Source Data file.

**mHtt interacts with ribosomes**. We next determined whether mHtt associates directly with ribosomes to regulate stalling by using multiple approaches. We isolated fractions from the ribosome profiles of control and HD-homo cells treated with vehicle or harringtonine (Fig. 6A). We found that wtHtt and mHtt both co-sedimented with the 40S, 60S, and 80S (MS) ribosomal subunits and with polysomes in sucrose gradients (Fig. 6A). Similarly, Fmrp, Rpl7, and Rpl35A (which are all known ribosome-binding proteins) also sedimented with the ribosomal subunits and polysome fractions (Fig. 6A). In the presence of harringtonine, which disassembles the polysomes[82,85], we found a clear increase in the monosome fraction in the ribosome profiling (Fig. 6A, arrowhead). Both wtHtt and mHtt re-sedimented from the higher-density polysome fractions (# 6, 7, or 8) to lower-density fractions (# 3, 4, or 5, asterisk) after harringtonine treatment. The known polysome-associated proteins, such as Fmrp, Rpl7, and Rpl35A, also re-sedimented to lower-density fractions (# 4, 5, or 6) after harringtonine treatment (Fig. 6A). The enhanced band intensity in fraction # 5 of harringtonine-

treated controls cells may be due to the overloading of monosome-accumulated proteins from the collected fractions. Besides, the shifting of ribosomal proteins Rpl7 and Rpl35A to the lower-density fractions appears very slight compared to Htt, which may be because Htt is less abundant on polysomes than ribosomal proteins. These results suggest that wtHtt and mHtt bind to translating polysomes.

Super-resolution stimulated emission depletion (STED) microscopy studies revealed that ~25–35% of wtHtt and mHtt clusters were in proximity (<300 nm) to the Rpl7 (Fig. 6B, the full image in Supplementary Fig. S5). We saw clear differences between the control and HD-homo cells when calculating the Manders colocalization coefficients (Fig. 6C). We also immunoprecipitated (IP) Htt using MAB2166 antibody and found that Rps6 and Rpl7 showed a stronger co-immunoprecipitation in Htt from HD-het and HD-homo cells than from control cells (Fig. 6D). As a positive control, we detected Caprin1, a previously known interactor of mHtt (Fig. 6D)[75,89]. Thus, the interaction with ribosomal proteins is stronger for mHtt than for wtHtt.

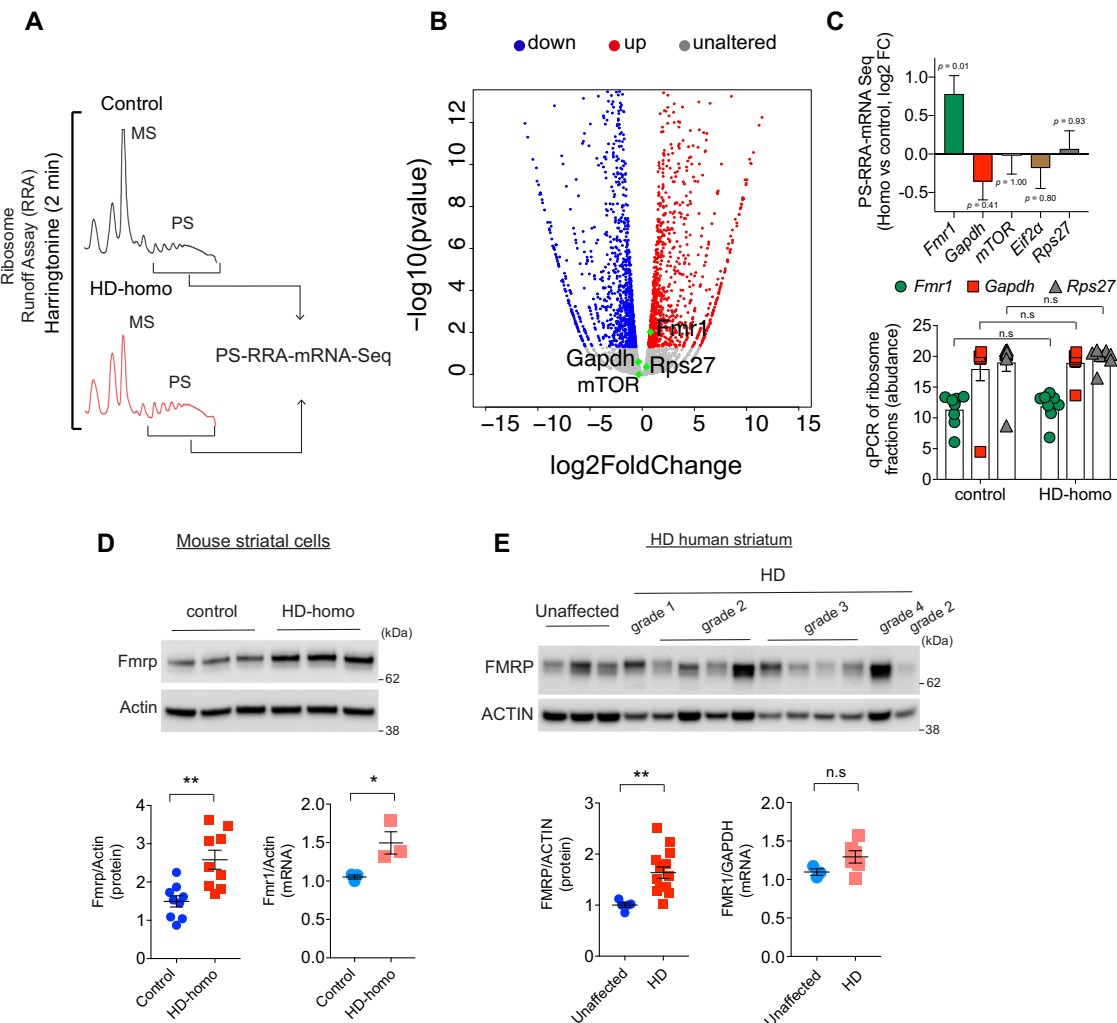

**Fig. 4 Fmrp is upregulated in HD. A** Representative polysome profiles of control and HD-homo cells after ribosome run-off assay using Harringtonine (2 μg/ml, 2 min). mRNA-Seq was performed using RNAs isolated from polysome containing fractions (arrows). **B** Volcano plot representing changes in gene expression levels of ribosome-bound mRNAs in HD-homo cells (vs control cells) after ribosome run-off assay from **A**. mRNA transcripts with absolute log2 fold change (log2FC) >0.3 and FDR-corrected $P$ value <0.05 are shown with red (up, increased polysome binding, log2FC >1), blue (down, decreased polysome binding, log2FC <1), gray (unaltered, no difference in polysome binding) and green (selected targets). Statistical testing was done using DESeq2 with a two-tailed Wald test and adjusted for multiple comparisons using the procedure of Benjamini–Hochberg. **C** A bar plot representing log2FC in the mRNA levels of some known translation regulating genes within polysome fractions obtained from HD-homo cells (comparing to controls) after ribosome run-off assay (upper panel, Data are log2FC and the standard error of the log2FC value from $n = 2$ independent experiments), and qPCR of the indicated targets in the purified ribosome fractions (bottom panel, Data are mean ± SEM, $n = 8$ independent experiments, n.s. not significant by two-way ANOVA, Bonferroni's multiple comparisons test). **D, E** Representative blots and mRNA levels and corresponding quantifications for Fmrp in mouse control and HD-homo striatal cells (Fmrp protein: $n = 9$ independent experiments, *Fmrp* mRNA: $n = 3$ independent experiments, **D**), and unaffected and human HD patient striatum (Fmrp protein: $n = 5$ unaffected, $n = 14$ HD, *FMRP* mRNA: $n = 3$ unaffected, $n = 6$ HD, **E**). Data are mean ± SEM, *$P < 0.05$, **$P < 0.01$ by two-tailed Student's $t$ test, n.s not significant. Exact $P$ values are reported in the Source Data file. Source data are provided as a Source Data file.

We asked whether mHtt could directly interact with translating ribosomes. We incubated purified recombinant human HTT proteins (GST-Exon 1-23Q or GST-Exon 1-51Q)[90] with polysomes isolated from Htt-depleted striatal cells (CRISPR/Cas9-Htt cells, Fig. 2A), and reran the isolated polysomes on a sucrose gradient (flow chart in Supplementary Fig. S6). We found stronger binding to the extracted polysomes with mHTT-51Q than with wtHTT-23Q (Fig. 6E), indicating that poly-Q expansion of HTT may increase HTT avidity toward ribosomes. However, we cannot rule out the possibility that the interaction of HTT with ribosomes may be enhanced in vivo by additional interactors and/or post-translational modifications.

To test this, we carried out whether mHTT bind to ribosomes in human HD-het fibroblasts, under acute amino acid starvation, which produces substantial transcriptional and translational changes. Human HD and healthy fibroblasts were either starved for amino acids (Krebs buffer) or starved and then stimulated with L-leucine (Leu), followed by immunoprecipitation with HTT IgG and control IgG and subjected to mass spectrometry (IP–LC-MS/MS) analysis. We found that HTT and mHTT both interacted with several 40S and 60S ribosomal proteins both in the starved and amino acid-stimulated conditions (Fig. 6F) in healthy and HD fibroblasts. STRING analysis found an enrichment of the biological process and components related to the translation and ribosome/RNA binding (Supplementary Fig. S7A, B). Full interactome data can be found in Supplementary Data 2 (see "Data availability" for protein database). Collectively, the above data demonstrate that Htt binds to translating ribosomes

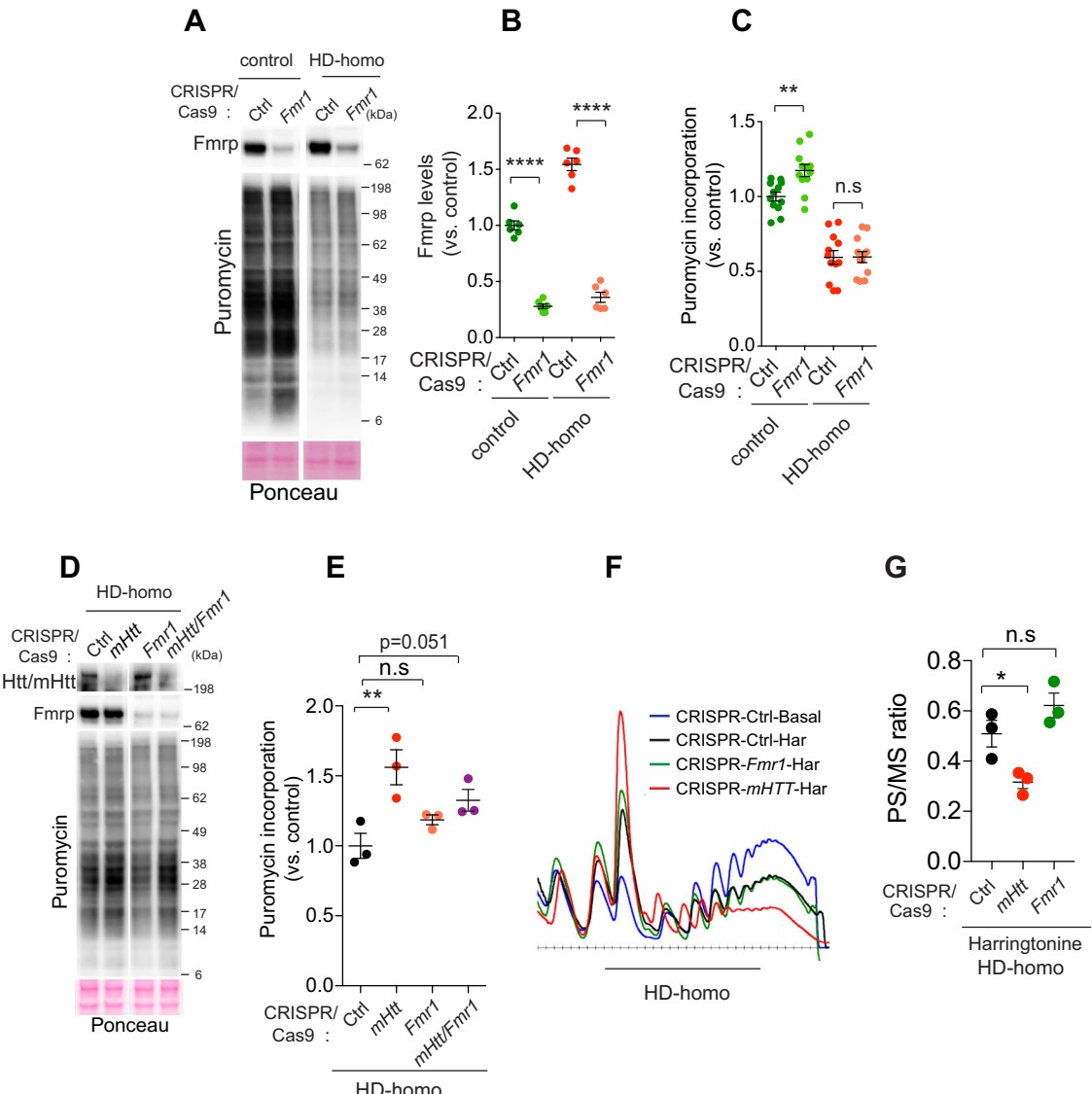

**Fig. 5 *Fmr1* depletion has no discernible effect on protein synthesis or ribosome stalling in HD cells. A** Representative immunoblots performed on CRISPR/Cas9 control (Ctrl) and *Fmr1*-depleted mouse striatal cells after puromycin metabolic labeling. **B, C** Quantification of blots from **A** showing FMRP levels (**B**), puromycin incorporation (**C**) in control (Ctrl) and *Fmr1*-depleted control, and HD-homo striatal cells. Data are mean ± SEM, (**B** *n* = 6; **C** *n* = 12 independent experiments), **\*\****P* < 0.01, \*\*\*\**P* < 0.0001, by one-way ANOVA followed by Bonferroni's multiple comparisons test, n.s not significant. **D** Representative immunoblots performed on *Htt or Fmr1* depleted, or *Htt/Fmr1* double-depleted HD-homo cells after puromycin metabolic labeling. **E** Quantification of the blots from **D**, showing puromycin incorporation. Data are mean ± SEM (*n* = 3 independent experiments), **\*\****P* < 0.01, by one-way ANOVA followed by Bonferroni's multiple comparisons test. *P* = 0.051 by two-tailed Student's *t* test. **F, G** Representative polysome profiles (**F**) obtained from *mHtt* or *Fmr1*-depleted mouse striatal cells before (basal) and after ribosome run-off assay using harringtonine (2 μg/ml, 3 min) and their corresponding quantification (**G**) of polysome to monosome (PS/MS) ratio (area under the curve). Data are mean ± SEM (*n* = 3 independent experiments), by one-way ANOVA followed by Tukey's multiple comparison test. n.s. not significant. Exact *P* values are reported in the Source Data file. Source data are provided as a Source Data file.

and ribosomal proteins in HD mouse striatal neuronal cells as well as human HD fibroblasts.

**Global ribosome profiling reveals diverse 5′ and 3′ ribosome occupancy on mRNA transcripts in HD cells.** The indications that mHtt binds and stalls ribosomes led us to posit that overall global ribosome occupancy may be altered in HD. We investigated this using a global ribosome profiling approach[91–94]. We treated cells with cycloheximide, as described previously[95,96], and isolated polysomes from three biological replicates of wild-type control striatal cells and HD-het and HD-homo mutant striatal cells. We conducted ribosome profiling to prepare ribosome footprints (Ribo-Seq) and

matching RNA (RNA-Seq) (Supplementary Fig. S8). Multiple quality control measures, such as principal component analysis (Fig. 7A) and Euclidian distance analyses (Supplementary Fig. S9A), showed that the replicates were very similar, with genotype as the principal source of differences between the control and the HD cells. Most of the differential ribosome occupancies (ribosome-protected fragments, RPF) were mapped to annotated protein-coding open-reading frames, with a 29-nt expected triplet nucleotide periodicity and ribosome occupancy at the start and stop codon for control, HD-het, and HD-homo (Fig. 7B and Supplementary Fig. S9B, C). Thus, we generated a high-quality Ribo-Seq library of control, HD-het, and HD-homo striatal cells for subsequent bioinformatics analysis.

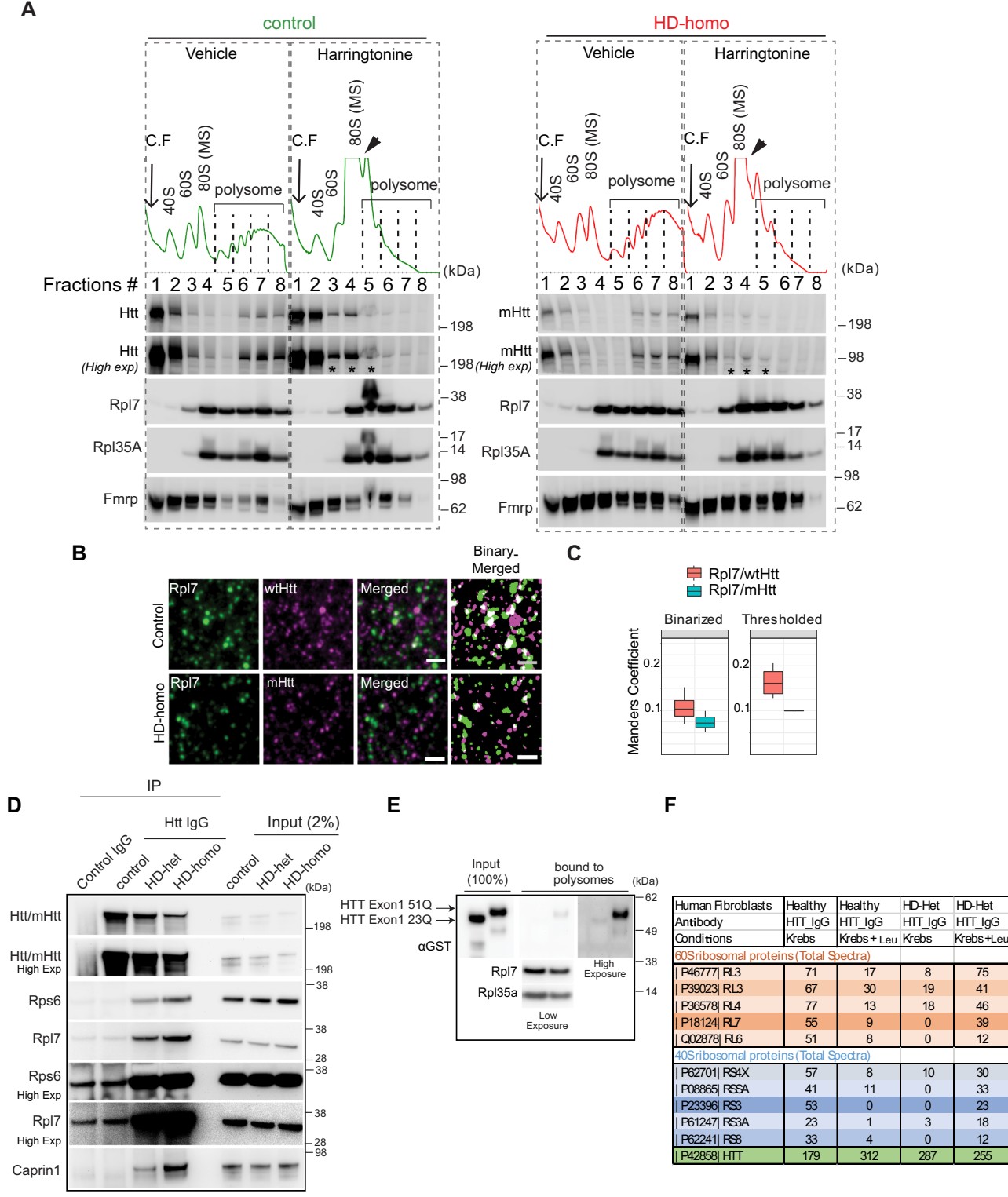

Because the data indicate ribosomes were stalled in HD cells, our goal was to identify mRNA targets with high RPF in HD. For this, we used Anota2Seq bioinformatics software[97], which applies partial and random variance models to allow the analysis of differential RPF (a.k.a. translation efficiency) between control and HD cells and estimation of the changes within each RNA source (RPF or mRNA)[98]. The following six categories of RPF/mRNA groups showing differential ribosome occupancy between control, HD-het, and HD-homo, striatal cells, were expected in our

analysis: (a) high RPF/similar mRNA (↑RPF/•mRNA), (b) similar RPF/low RNA (•RPF/↓mRNA), (c) low RPF/similar mRNA (↓RPF/•mRNA), (d) similar RPF/high mRNA (•RPF/↑mRNA), (e) high RPF/high mRNA (↑RPF/↑mRNA), and (f) low RPF/low mRNA (↓RPF/↓mRNA) (Supplementary Fig. S10A).

We first confirmed that the Anota2Seq algorithms were able to separate the ribosome occupancy of each category from HD-het and HD-homo cells versus controls (Fig. 7C and Supplementary Data 3). As shown in the scatterplot (Fig. 7D), we found that the

**Fig. 6 Huntingtin interacts with ribosomes. A** Representative polysome profile and immunoblots with indicated proteins in the fractions from the gradients of control and HD-homo cells treated with vehicle or Harringtonine (2 μg/ml, 30 min) ($n = 3$ independent experiments). **B** Representative images of STED microscopy on mouse striatal cells using antibodies against Rpl7 (green) and Htt (magenta) counterstained with the nuclear marker DAPI (see the full image in Supplementary Fig. S5) Scale bar = 500 nm, $n = 4$ for control cells, $n = 3$ for HD-homo cells. **C** The Manders' colocalization coefficients were calculated for the ratio of FarRed-label colocalizing with Red-label, for both control and homo conditions, using background subtracted (thresholded) and binarized STED images from **B**. The central line details the median (50%), whereas the upper and lower boundaries of the box (hinges) represent the first (25%) and third (75%) quartile of the data. The whiskers extend up to the largest and down to the smallest value, all of which were inside 1.5*IQR the interquartile range, or the distance between the first and third quartiles, $n = 4$ for control cells, $n = 3$ for HD-homo cells. **D** Representative immunoblots showing co-immunoprecipitation of wHtt/mHtt (MAB2166) with indicated proteins, and input, $n = 4$ independent experiments. **E** Representative immunoblots on inputs and outputs of in vitro ribosome-binding assays using recombinant GST-exon 1 HTT 23Q and GST-exon 1 HTT 51Q proteins and isolated ribosomes from mouse *Htt*-depleted striatal cells ($n = 3$ independent experiments, see Supplementary Fig. S6 for the experimental diagram and corresponding polysome profiles). **F** Proteomics analysis on samples prepared from immunoprecipitation experiments on human (healthy controls 17Q and HD patient 69Q-het) fibroblasts in Krebs or Krebs + 3 mM Leu (four control IgG and four HTT IgG, $n = 1$ per group).

36,411 genes included 3840 mRNA targets in the HD-homo cells and 6670 mRNA targets in the HD-het cells, with significantly changed ribosome occupancy (RPF/mRNA) compared to the controls ($P$ value <0.05) (Supplementary Data 4 and **5**). This differential RPF change in HD could also be due to transcript-level changes consistent with the transcriptional role of mHtt[99]. Gene ontology (GO), using the Ingenuity Pathway Analysis (IPA) for each category, showed that the genes belong to diverse signaling pathways, including IL-10 signaling, Alzheimer's disease signaling, AMPK signaling, cAMP signaling, Rho GTPase signaling, and the matrix metalloprotease, EIF2, and mTOR pathways (Supplementary Fig. S10B). Therefore, we were able to ensure our Ribo-Seq data to detect robust differences in the RPF in HD cells compared to the control.

We next sought to identify the overall ribosome pauses on mRNA transcripts in HD using PausePred software[100]. Linear regression analysis using the mean of triplicates allowed a determination of the number of cases with a 5′ ribosomal shift and a 3′ ribosomal shift in the HD and control cells. The HD-homo cells showed more genes with 5′ ribosome occupancy (5′ = 790; 3′ = 138), whereas HD-het cells showed more genes with 3′ occupancy (5′ = 74; 3′ = 1685) (Fig. 8A, Supplementary Data 6 and 7). For example, the mRNA of *B-myc*, a brain-expressed myelocytomatosis oncogene[101], showed high ribosome occupancy toward the 5′ end in HD-homo (Fig. 8B, arrow). By contrast, *mt-Nd4I* mRNA, a subunit of NADH dehydrogenase (ubiquinone), displayed high ribosome occupancy toward the 3′ end (Fig. 8B, arrowhead). These data indicate that the center of the ribosome density is significantly shifted on selected targets in HD cells compared to normal cells.

The PausePred software can also be used to analyze the global single-codon pauses[100]. However, although we observed numerous single-codon pauses between control and HD cells (Supplementary Fig. S11), the high level of variation within the replicates (Supplementary Fig. S12) precluded any comparison of pauses across groups. These variations within the replicates may also indicate that a ribosomal pause on a given codon is not a static but a dynamic event in striatal neuronal cells. We therefore selected the codons that showed HD-exclusive pauses compared to control cells. This approach identified ~165 targets in the HD-homo cells and ~125 targets in the HD-het cells that showed one or more codon-specific pauses (Supplementary Data 8 and 9). Most of the targets showed ribosome pauses toward the 5′ end as well as the 3′ end of the coding mRNA in HD cells compared to controls (Fig. 8C). Although we can draw no clear conclusion whether the observed pause signal was a result of pausing or simply due to differences in the transcript, we found that most of the single-codon paused transcripts (~130) were enriched in the top PS-bound mRNA list in PS-RRA-mRNA-Seq of the HD-homo cells (Fig. 8D) (Supplementary Data 10).

We next examined the ribosome profiles of some of the targets identified in both Ribo-Seq and PS-RRA-mRNA-Seq and their protein expressions (depending on the availability of antibodies). We combined the uploaded profiles, as a track hub in the University of California Santa Cruz (UCSC) Genome Browser 42, from the triplicate experiments and overlaid selected RPF (green) normalized to corresponding mRNA (orange). We also estimated the ribosome occupancy as the ratio between CDS RPF abundance and mRNA abundance and for the indicated gene (RPF/mRNA) from the raw read counts from the UCSC browser (Fig. 9A–F, bar graphs). For example, *Mfsd10*, an ion transporter, showed a single-codon pause at position 1673 (GAG) and enhanced RPF/mRNA reads in the HD-het and HD-homo cells (Fig. 9A, arrow). By contrast, *Acan*, a proteoglycan consisting of seventeen exons, showed more or less uniformly distributed ribosome occupancy in control and HD-het cells. Its single-codon pause was at 2515 (TGG), and its RPF was concentrated predominantly toward the 5′ of the mRNA transcript in HD-homo cells (Fig. 9B, arrow). *Ppbp*, platelet-derived growth factor, which belongs to the CXC chemokine family, consists of three exons and showed high RPF concentrated on exon 2 in HD-het and HD-homo compared to control (Fig. 9C, arrow). *Mgp*, N-methylpurine-DNA glycosylase, showed a single-codon pause at 100 (AGC) and high RPF reads in HD-het and HD-homo cells compared to control cells, with RPF tilted more toward 3′ in the HD-het cells (Fig. 9D, arrow). *Phf11d*, a protein containing a PHD (plant homeodomain) type zinc finger, has 11 exons and showed high RPF in HD-homo (18-fold) compared to control cells (Fig. 9E, arrow). mTOR, a widely studied kinase, shows a high RPF at the 5′ end of the mRNAs but it gradually tapered down along the 58 exons in all groups (Fig. 9F).

Therefore, RPF/mRNA analysis showed overall high RPF in Mfsd10, Ppbp, Mgp, and Phf11d targets in HD cells compared to control cells. Apart from Acan, the rest of the targets showed diminished protein levels by western blotting in HD cells compared to control, but the mTOR protein levels were not significantly altered (Fig. 9G). Thus, despite a high RPF density, certain selected targets showed diminished or unaltered protein levels in HD cells compared to control cells, suggesting a possible translational regulation including ribosome stalling and/or post-translational mechanisms of protein stability in HD.

We have recently evaluated whether paused targets play any role in HD. We reported that cGAS, a DNA sensor[102] (Fig. 8D, arrow), showed high ribosome occupancy at its exon 1 and a pause at 171 (CCG) and 172 (CGT) in HD cells[103] and that the cGAS protein is highly upregulated in HD cells, HD mouse models, and human patient tissues. We also showed that cGAS promotes an inflammatory and autophagic response in HD cells[103]. A full UCSC genome browser link for the global RPF averaged for all three replicates for control, HD-het, and HD-

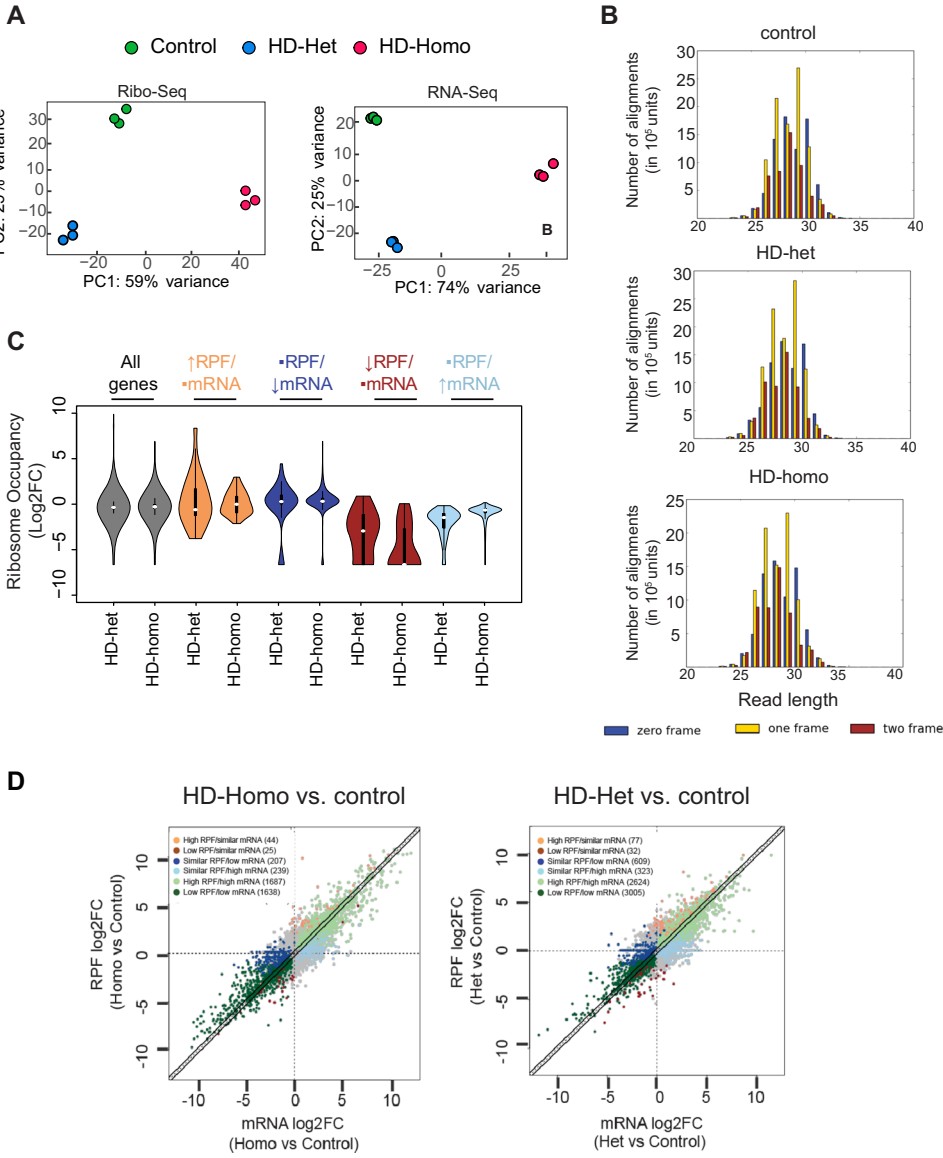

**Fig. 7 Global ribosome profiling reveals diverse ribosome occupancy on mRNA transcripts in HD cells. A** Principal component analysis using Ribo-Seq and RNA-Seq data obtained from control, HD-het, and HD-homo striatal cells. **B** Representative graphs showing triplet periodicity plots generated for protein-coding regions of Ribo-Seq data obtained from control, HD-het and HD-homo striatal cells. **C** Violin plots showing the distribution of ribosome occupancy changes (calculated by the number of Ribo-Seq reads divided by mRNA-Seq reads for each gene, Log2FC) in HD-homo and HD-het cells (compared to control cells). In the middle of each density curve is a small boxplot, with the rectangle showing the ends of the first and third quartiles and central dot the median ($n = 3$ independent experiments). **D** Scatter plots of expression changes of Ribo-Seq vs mRNA-Seq data in control, HD-het, and HD-homo striatal cells. Using Anota2seq, changes in Ribo-Seq are classified into six groups (see Supplementary Fig. S10A). The numbers of genes in each group are shown at the top left corner of each plot. The total number of genes: 36411, absolute fold change (FC) > 2, nominal $P < 0.05$, and false discovery rate (FDR) = 0.15. All statistical tests within the anota2seq package are two-tailed.

homo cells can be found at the UCSC genome browser (see "Data availability" for the link).

Thus, our Ribo-Seq analysis revealed robust differences in the RPF/mRNA, 5′, and 3′ distribution of ribosomes and single-codon pauses on selected mRNA transcripts. This study also provides a rich source of potential unknown targets for drug discovery in HD research.

## Discussion
Despite many efforts, no effective therapies are yet available that are directed toward inhibiting the abnormal functions of mHtt in HD. Therefore, delineating the mechanisms become important for developing effective therapies against HD. Based on the present study findings, our model indicates that wtHtt physiologically inhibits protein synthesis by inhibiting the speed of ribosomal translocation. This function is exacerbated by mHtt, which further impedes protein synthesis and aberrantly slows down ribosomal translocation (Fig. 9H). This model is supported by multiple pieces of evidence from the present study: (i) an enhanced level of polysome-bound mRNA transcripts but a diminished level of protein synthesis in HD cells, (ii) increased protein synthesis as well as increased speed of ribosome runoff by depletion of mHtt, (iii) direct inhibition of protein synthesis in vitro by mHtt, (iv) an association between mHtt and translating ribosomes, and (v) widespread alterations in the ribosome occupancy in HD cells compared to control cells.

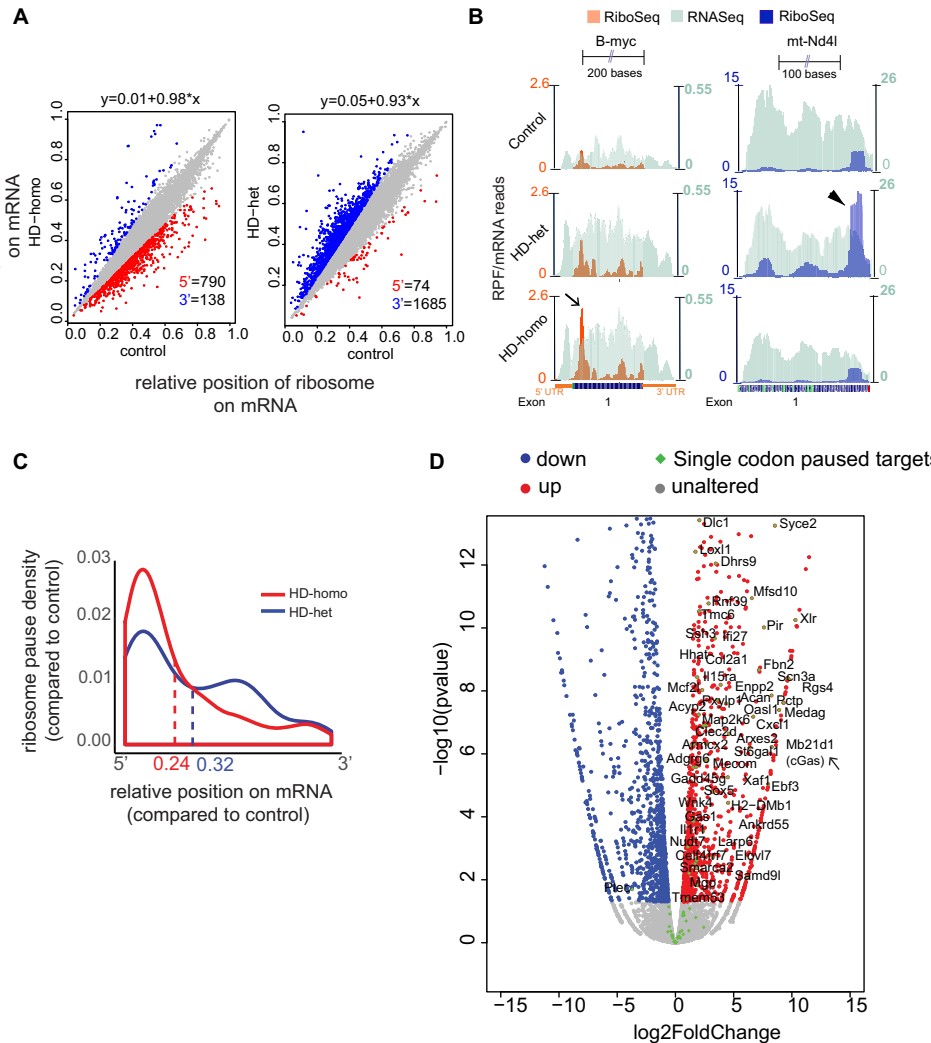

**Fig. 8 Analysis of 5′- and 3′-end ribosome occupancy and single-codon pauses in HD cells. A** Each dot represents the mean center ribosome density score of the same transcript in HD cells vs control cells (n = 3). Red dots represent transcripts with higher 5′ ribosome occupancy in HD cells vs control cells. Blue dots represent transcripts with higher 3′ ribosome occupancy in HD cells vs controls. The axis indicates the relative position of the ribosome on the mRNA transcript. **B** Graphs showing the overlay of Ribo-Seq (RPF)/mRNA-Seq reads for *B-myc* and *mt-ND4I*, gene obtained from UCSC browser. **C** Pause density plot showing the distribution of common paused codons (found among three replicates of HD cells versus control cells) over their corresponding mRNAs. Numbers showed at the bottom represent the average of relative positions of paused codons in HD-homo (red) and HD-het (blue) cells. **D** Volcano plot from Fig. 4B representing changes in gene expression levels of ribosome-bound mRNAs in HD-homo cells (vs control cells) after ribosome run-off assay, showing single-codon paused targets (green). mRNA transcripts with absolute log2 fold change >0.3 and FDR-corrected *P* value <0.05 are shown with red (up, increased polysome binding, log2FC > 1), blue (down, decreased polysome binding, log2FC < 1), gray (unaltered, no difference in polysome binding). Statistical testing was done using DESeq2 with a two-tailed Wald test and adjusted for multiple comparisons using the procedure of Benjamini–Hochberg.

Surprisingly, despite an approximately two-fold increase in the levels of Fmrp, a previously known regulator of ribosome stalling and mRNA translation[18,104–107], depletion of Fmrp did not significantly affect ribosome stalling or protein synthesis in HD cells. Previous studies have shown a biochemical association of Fmrp and Htt, as *Htt* mRNA is a target for Fmrp binding in the brain[16], and Fmrp binds with 3′ UTR of *Htt* mRNA[108]. A recent study showed that Fmrp deficiency leads to reduced *Htt* mRNA expression and that Htt mediates the Fmrp regulation of mitochondrial fusion and dendritic maturation[109]. Thus, Htt and Fmrp may act together in a cell-type-specific manner and have yet undiscovered biological functions in the brain. Although Fmrp upregulation did not affect translation in HD cultured striatal cells, we cannot rule out its role in HD pathogenesis. This notion is supported by evidence suggesting that deletion of Fmrp (dfmr1) suppresses the mHTT-mediated toxicity in the

Drosophila model of HD[110]. In the HD brain, hypothetically, Fmrp may regulate the translation of selected mRNA transcripts involved in synaptic functions[104,111,112]. In addition, Fmrp upregulation may impact noncanonical functions, such as autophagy regulation in HD[113,114]. Thus, our study demonstrates a previously unknown link for mHtt and Fmrp and its relevance to HD pathogenesis requires further studies.

Our results demonstrating a role for Htt in protein synthesis in HD is consistent with the findings of previous studies[73,75–78]. These earlier studies showed that full-length wtHtt or mHtt interacted with ribosomal proteins in the brain and inhibited cap-dependent translation of a reporter mRNA in Hela cells[75]. Similarly, the expression of the fragment of the first 17 amino acids of Htt fused to a poly(Q) tract of 103 (Htt103Q) in yeast cells and Drosophila cells diminished the expression of genes involved in rRNA metabolism and ribosome biogenesis, while

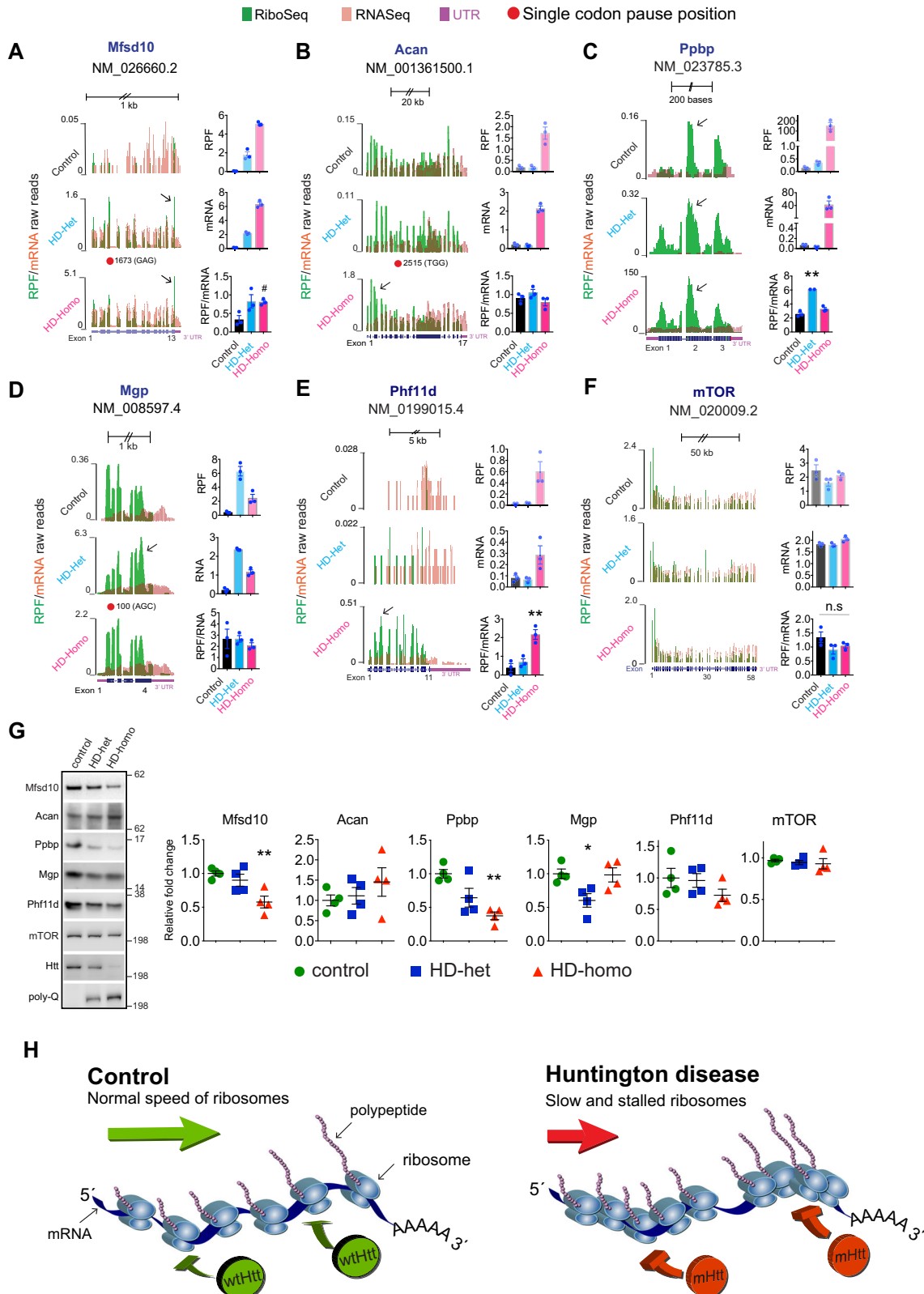

also suppressing protein synthesis[73,76]. Protein/gene expression profiling has revealed diminished protein levels, altered expression of genes encoding ribosomal proteins, and RNA processing proteins in various HD models[51,70,71,74,115,116], indicating a strong translational role for mHtt. But the mechanisms of how mHtt might promote translational abnormalities were not revealed by these previous works.

Our data show both wtHtt and mHtt appear to act as inhibitors of translation elongation, we predict that they can elicit gain or loss of function activities toward protein synthesis, depending upon how much (levels) and what (with or without mHtt) is present on the ribosomes as well as where Htt/mHtt is present (localization). In support of this notion, our super-resolution Ribo-Seq data revealed markedly different RPF/mRNA profiles

**Fig. 9 Ribosome stalling was confirmed in several genes by measuring protein levels. A–F** Representative graphs showing the overlay of Ribo-Seq (RPF)/ mRNA-Seq reads for indicated genes obtained from UCSC browser. Bar graphs indicate RPF abundance, mRNA abundance, and ribosome occupancy as the ratio between CDS of RPF and mRNA for the indicated gene (RPF/mRNA) from the raw read counts from the UCSC browser. Data mean ± SEM (n = 3 independent experiments, #P < 0.05 by two-tailed Student's t test, **P < 0.01 by one-way ANOVA followed by Tukey's multiple comparison test. n.s. not significant. **G** Representative immunoblots showing the protein expression levels of indicated genes in **A–F** and the corresponding quantifications. Data mean ± SEM, n = 4 independent experiments, *P < 0.05 and **P < 0.01 by one-way ANOVA followed by Tukey's multiple comparison tests. Exact P values are reported in the Source Data file. Source data are provided as a Source Data file. **H** Model showing wtHtt bind to ribosomes and physiologically inhibits the translocation of ribosomes on mRNA, a normal function that is further enhanced by mHtt leading to slower movement and stalling of ribosomes. Note stoichiometry of wtHtt/mHtt binding to ribosomes is currently unknown.

between HD-homo cells and the control and between HD-het and the control cells (Fig. 7D). Previously, we showed that mHtt activates amino acid-induced Rheb–mTORC1 signaling, which promotes the initiation of mRNAs containing the 5′-terminal oligopyrimidine (5′ TOP) motif[117,118]. We also showed Rheb can inhibit protein synthesis by activating the PERK–eIF2α signaling cascade[119]. Interestingly, a recent study demonstrated enhanced protein synthesis in the brains of R6/1 transgenic and Hdh$^{Q7/Q111}$ knock-in mice, while Q175FDN mice showed diminished levels of soluble proteins[77,120]. Thus, mHtt may modulate protein synthesis depending upon the interacting partners, nutritional status of the cells, and the changes in the intact brain environment in vivo.

Previous elegant studies have suggested multiple players in the regulation of ribosome stalling, including secondary RNA structures, 3′ UTR, codon usage, or the availability of charged tRNAs, ribosomal-binding proteins, and/or nascent polypeptide chains, which may all contribute to ribosomal pauses[1–4]. Whether similar mechanisms operate in striatal neuronal cells, and whether mHtt recruits one or more of these players to stall ribosomes, remains unknown. Htt consists of 28–32 HEAT [Huntingtin, elongation factor 3 (eEF3) 1, protein phosphatase 2A (PP2A) 2, and the yeast PI3-kinase TOR1] repeats that span the entire protein. Many translation regulators, such as eEF3, eIF4Gs, p97DAP5, GCN1, and mTOR, contain HEAT repeats. These repeats are tandem repeats of an alpha-helical hairpin that form a superhelical structure with hydrophobic cores; this structure is predicted to interact with other proteins and/or nucleic acids[121,122]. Structural analysis has shown that yeast eEF3 HEAT (13) repeats interact with rRNA and the ribosomal proteins of the small ribosomal subunit, and this is proposed to play a role in the translocation of aminoacyl-tRNA from the A site to the P site on the ribosome[123–125]. Similarly, yeast eIF4G HEAT (four) repeats have been shown to interact with eIF4A and to stimulate its ATPase activity to facilitate 43S preinitiation complex recruitment and movement along the mRNA in a 5′ to 3′ direction[126]. Thus, mHtt may interact with ribosomes via HEAT repeats to modulate ribosome stalling.

Short fragments and aggregated forms of Htt may also differentially influence ribosome stalling. Htt is proteolytically cleaved by several proteases, resulting in smaller N-terminal poly-Q containing exon 1 and non-poly-Q Htt C-terminal fragments, which are considered important for disease pathogenesis[54,127,128]. Although we found FL-mHtt present on the polysomes and interacts with ribosomal proteins, previous work has shown that exon 1 aggregates could also be associated with ribosomes[129,130]. Therefore, different Htt fragments (e.g., poly-Q exon 1 and HEAT repeat domains) and aggregated forms[54,63,131] may differentially bind to the translating ribosomes. For example, mHtt aggregates are sequestered by the cytoplasmic polyadenylation element-binding protein/Orb2[76], a translational regulator[132,133], whose mRNA shows an enhanced ribosome occupancy in HD (see UCSC browser link), indicating Orb2 may be involved in ribosome stalling in HD. Notably, the association of protein aggregates with ribosomes, as found with various neurodegenerative disease-linked proteins, may have a considerable impact on the mRNA translational machinery[134–136]. Further identifying the nature and stoichiometry of Htt fragment binding to the ribosomes and its binding partners may yield molecular insights into dynamic protein synthesis regulation in HD cells.

One question that remains is whether or how mHtt-mediated ribosome stalling influences striatal-specific damage in HD[137,138]. We previously showed that striatal-enriched Rhes promotes HD toxicity in cell and animal models[139,140] by enhancing the SUMOylation of mHtt and increasing its solubility[139,141]. Several independent studies support a toxic role for Rhes in various HD models[139,140,142–148]. Recently, we demonstrated that Rhes promotes cell-to-cell transport of mHtt via tunneling nanotube (TNT)-like protrusions[149]. Rhes also activates mTORC1 signaling[150], a known mediator of mRNA translation[151,152], but how this may influence ribosome stalling in HD remains to be determined. We propose that a Rhes-mediated enhancement of mHtt solubility may further worsen ribosome stalling in the striatum and interfere with Rhes-mediated intercellular communication via the TNT-like protrusions; other cellular and inflammatory processes may also contribute to ribosome stalling and selective neurodegeneration in HD[68,103,137,138,153].

Taken together, our findings indicate that mHtt suppresses protein synthesis via ribosome stalling potentially during translation elongation. This altered function may mediate progressive and widespread development of HD-related behavioral and pathological symptoms. Developing drugs that interfere with mHtt-mediated mechanisms of ribosome stalling may prevent or slow down the progression of HD.

## Methods

**Chemicals and cell culture**. Chemicals and reagents were mainly purchased from Sigma. Mouse striatal cells (ST*Hdh*) expressing knock-in wild-type HTT$^{exon1}$ with 7 glutamine (Q) repeats (control; ST*Hdh*$^{Q7/Q7}$) or expressing knock-in mutant human HTT$^{exon1}$ with 111 glutamine repeats (HD-het; ST*Hdh*$^{Q7/Q111}$, and HD-homo; ST*Hdh*$^{Q111/Q111}$)[80] were purchased from Coriell Institute for Medical Research (Camden, NJ, USA) and cultured in Dulbecco's modified Eagle's medium, high glucose, GlutaMAX supplement (DMEM) (Thermo Fisher Scientific) supplemented with 10% fetal bovine serum (FBS) (Thermo Fisher Scientific), 1% penicillin–streptomycin 5% $CO_2$, at 33 °C, as described before[118]. HD patient-derived fibroblast cell lines GM04281 (wild-type HTT allele/17 CAG repeats, mutant HTT allele/69 CAG repeats) and normal human fibroblast cell line GM07492 were obtained from Coriell Institute for Medical Research (Camden, NJ, USA). Cells were maintained at 37 °C and 5% $CO_2$ in DMEM, high glucose, GlutaMAX supplement supplemented with 10% FBS, 1% penicillin–streptomycin, and 1% MEM nonessential amino acids (Thermo Fisher Scientific).

**Antibodies**. The following commercial antibodies were used: Huntingtin (MAB2166, 1:3000) and puromycin (MABE343, 1:10,000) antibodies were obtained from Millipore-Sigma. Anti-polyglutamine (poly-Q) antibody (P1874, 1:5000) was from Sigma. Actin (sc47778, 1:20,000) and GST-horseradish peroxidase (HRP, sc138 HRP, 1:10,000) antibodies were from Santa Cruz Biotechnology. RPL7 (IHC-00455, 1:10,000), RPL35A (A305-106A, 1:10,000) and Caprin1 (A303-881A, 1:1000) from Bethyl Laboratories. mTOR (2972, 1:3000), FMRP (4317, 1:1500), and S6 (2217, 1:10,000), and normal mouse IgG (5415, 1:2500) were from Cell Signaling Technology. Mfsd (10 11518-1-527 AP, 1:1000), Acan (13880-1-AP,

1:1000), Ppbp (13313-1-AP, 1:1000), Mgp (10734-1-AP, 1:1000), and Phf11d (10898-1-AP, 1:1000) were from Proteintech. HRP-conjugated secondary antibodies were from Jackson ImmunoResearch Inc. HRP-conjugated secondary antibodies (115-035-146 (goat anti-mouse), 1:10,000; or 111-035-144 (goat anti-rabbit), 1:10,000) were from Jackson ImmunoResearch Inc. For Immunostaining Huntingtin (1:100, MAB2166), Rpl7 (1:50; IHC-00455) were used. Secondary antibodies for STED microscopy were anti-mouse STAR 635p (1:400) and anti-rabbit Alexa 594 (1:400) that were self-coupled in-house and the company that produced them is Abberior, Göttingen, Germany.

**Deletion of Hdh (Htt) and Fmr1 (Fmrp) genes using CRISPR/Cas9.** GFP expressing CRISPR/Cas9 knockout plasmids for mouse Htt (sc-420825) and Fmr1 (sc-420392) or CRISPR/Cas9 control plasmid (sc-418922) were purchased from Santa Cruz Biotechnology. Mouse striatal cells were transfected with knockout plasmids by PolyFect (Qiagen) following recommendations by the company. The transfected cells were harvested 48 h later, washed, and resuspended in the sorting buffer (containing 1× phosphate-buffered saline (Ca/Mg++ free), 2.5 mM EDTA, 25 mM HEPES pH 7.0, 1% fetal bovine serum, and 10 unit/ml DNase1). The GFP expressing cells were sorted using a BD biosciences Aria FACs machine. The sorted cells were plated and maintained in DMEM high glucose containing 10% FBS at 33 °C.

**Immunostaining of the cells and STED image acquisition.** Mouse striatal cells were plated in 12-well plates containing glass coverslips. The day after, the cells were washed with PBS and fixed using 4% PFA in PBS, then permeabilized in 0.1% Triton X-100 in PBS and blocked using 5% normal donkey serum/ 1% BSA/ 0.1% Tween in PBS. The cells were incubated with primary antibodies (Htt 1:100, MAB2166, Rpl7 1:50; IHC-00455) in blocking buffer at 4 °C overnight. The day after, the cells were washed and incubated with secondary antibodies and DAPI in 1% BSA/ 0.1% Tween in PBS for 1 h at room temperature. The stimulated emission depletion (STED) microscopy imaging was performed using a multicolor Expert-Line STED nanoscope (Abberior Instruments GmbH), using the 775-nm pulsed STED laser in combination with the 561-nm and 640-nm pulsed excitation lasers, as well as the 405-nm excitation for diffraction-limited imaging of DAPI. All images were recorded simultaneously with diffraction-limited (i.e., confocal) and with 2D-STED enhanced resolution, recording each color channel and resolution in a line-interleaved manner. Pixel sizes ($x$, $y$) were 20-nm × 20-nm for both STED and confocal images, with typical pixel dwell times of 10 μs for confocal and 30—45 μs for STED images. The images were recorded with a ×100 oil immersion objective lens (NA 1.4), using the QUAD beam scanner, utilizing the Imspector software package (Max-Planck Innovation).

**Image analysis.** For the colocalization analysis, raw STED images from Imspector were imported into FIJI and were processed as follows. First, raw images were gently smoothed with a one-pixel Gaussian filter. Next, an appropriate background level was determined individually for each image, striking a careful balance between being able to distinguish individual clusters in proximity without losing any of the dimmer features. This background value was then subtracted from the images. For some parts of the analysis, the images were then binarized. Regions coinciding with the cell nucleus were excluded from the colocalization analysis, as were smaller regions coinciding with any obvious staining artifacts. The colocalization analysis itself was performed using the ImageJ plugin JACoP, utilizing both the pixel-based Manders coefficient analysis and the object-based methods. The data is presented as boxplots that were generated using R Statistics (www.r-project.org).

**Puromycin metabolic labeling and immunoblotting.** Mouse striatal cells were plated at a confluency of about 60–70%. The day after the cells were incubated with puromycin (20 μM final concentration) for 5 min, as described[81]. Then cells were rinsed with cold PBS and immediately were lysed in RIPA buffer containing protease inhibitors. Equal proteins were used to run western blotting experiments. The puromycin incorporation was normalized to the ponceau S staining.

**Immunoprecipitation in striatal cells.** Control, HD-Het, and HD-Homo striatal cells (2 × 10⁶) were plated in 10-cm dishes, and next day were lysed in immuno-precipitation (IP) buffer (15 mM HEPES (pH 7.3), 7.5 mM MgCl₂, 100 mM KCl, 1.0% Triton X-100, 1 mM dithiothreitol (DTT), EDTA-free protease inhibitor cocktail (Roche), RNasin (40U/μl, Promega)). The lysates were run several times through a 26-gauge needle in IP buffer and incubated on ice for 15 min and centrifuged 20,000×g for 15 min. Protein estimation in the lysate supernatant was done using a bicinchoninic acid (BCA) method, a concentration (1 mg/ml) of protein lysates was precleared with 40 μl of protein A/G beads for 1 h, the supernatant was incubated for 1 h at 4 °C in HTT IgG (MAB2166) or control IgG, and then 60 μl protein A/G beads were added and incubated overnight at 4 °C. After 12 h, the beads were washed five times with IP buffer (without RNasin/protease inhibitor), and the protein samples were eluted with 30 μl of 2× lithium dodecyl sulfate (LDS) containing +1.5% β-mercaptoethanol, separated on NuPAGE 4–12% Bis–Tris gel (Thermo Fisher Scientific), transferred to polyvinylidene difluoride membranes, and probed with the indicated antibodies. HRP-conjugated secondary antibodies (Jackson ImmunoResearch Inc.) were probed to detect bound primary

IgG with a chemiluminescence imager (Alpha Innotech) using enhanced chemi-luminescence from WesternBright Quantum (Advansta).

**Immunoprecipitation in human fibroblasts and LC-MS/MS.** Human healthy and HD fibroblasts were plated in 10-cm dishes and the next day the medium was changed to Krebs buffer medium (20 mM HEPES pH 7.4, glucose (4.5 g/liter), 118 mM NaCl, 4.6 mM KCl, 1 mM MgCl₂.6H₂O, 12 mM NaHCO₃, 0.5 mM CaCl₂, 0.2% (w/v) bovine serum albumin (BSA)) devoid of serum and amino acids for 1 h to simulate full starvation conditions. For the stimulation conditions, cells were stimulated for 15 min with 3 mM L-Leucine (Leu). Cells were lysed and proceeded for IP for HTT as mentioned above for the striatal cells using HTT IgG (MAB2166) and control IgG. After running the IP samples in electrophoresis, the samples were subjected to IP–LC-MS/MS as described previously[154] for the analysis of HTT interactors.

Eight protein samples named H1, H2, H3, H4, H5, H6, H7, and H8 (details in Supplementary Table 1) and 4 mg of BSA control were subjected in parallel to SDS-PAGE at 120 V for 12 min. The gel was Coomassie-stained for 1 h at room temperature with shaking, followed by de-staining in water overnight. The gel bands were cut, in-gel treated with 25 mM DTT followed by 55 mM iodoacetamide, and subjected to trypsin digestion with ProteaseMax Surfactant Trypsin Enhancer for 1 h at 50 °C. The peptide pools were acidified and desalted through Zip-Tip μC18 tip columns. Prior to mass spectrometry analysis, the samples were reconstructed in 5 μl of 0.1% formic acid and 5 μl were loaded into the system. Each sample was analyzed by an Orbitrap Fusion Tribrid Mass Spectrometer (Thermo Fisher Scientific) coupled to an EASY-nLC 1000 system. Peptides were online eluted on an analytical RP column (0.075 × 150 mm Acclaim PepMap RLSC nano Viper, Thermo Fisher Scientific), operating at 300 nl/min using the following gradient: 5–25% B for 90 mins, 25–44% B for 30 min, 44–80% B in 10 s, 80% for 5 min, 80–5% B in 10 s, and 5% B for 40 min (solvent A: 0.1% formic acid (v/v); solvent B: 0.1% formic acid (v/v), 80% CH₃CN (v/v) (Fisher Scientific)). The Orbitrap Fusion was operated in a data-dependent MS/MS mode using top speed precursor selection detected in a survey scan from 380 to 1400 mass/charge ratio ($m/z$) performed at 120 K resolution. Tandem MS was performed by higher-energy collisional dissociation fragmentation with a normalized collision energy of 30.0%. Protein identification was carried out using Sequest algorithms (Proteome Discoverer v1.4, Thermo Scientific), allowing oxidation (Met) and deamination (Q) as variable modifications. Other settings included carbamidomethylation of Cys as a fixed modification, three missed cleavages, and mass tolerance of 10 ppm and 0.02 Da for precursor and fragment ions, respectively. MS/MS raw files were searched against a Uniprot human database. The FDRs of peptide identifications were calculated from the search results against a reverse sequence database; 1% FDR, was used as a criterion for peptide identification (the list of peptide identification is presented in data files Supplementary Data 2). Scaffold (version Scaffold_4.8.8, Proteome Software Inc., Portland, OR) was used to validate MS/MS-based peptide and protein identifications. The complete dataset from the analysis of the interactors (raw files, identification data, and data analysis files) can be obtained via ProteomeXchange with identifier PXD017115 at http://www.proteomexchange.org/.

**Ribosome isolation and in vitro binding assay.** For each assay, four 15-cm plates of mouse striatal cells were used. Briefly, the cells were incubated with 100 μg/ml cycloheximide (CHX) for around 10 min at 37 °C, then harvested, spin down, and washed once with cold PBS containing CHX. The cells were lysed in the lysis buffer containing 20 mM HEPES pH 7.3, 150 mM KCl, 10 mM MgCl₂, 2 mM DTT, 100 μg/ml CHX, 0.5% v/v Triton X-100, 20 U/ml RNasin and EDTA-free protease inhibitor cocktail (Roche). The cell lysates were loaded on 10–50% sucrose gradients and centrifuged at 280,000 ×g (SW41Ti rotor) at 4 °C for 2 h. Gradients were fractionated using a gradient fractionator and UA-6 detector, 254-nm filter (ISCO/BRANDEL). The fractions containing monosome and polysomes were collected and transferred to a 50-mL centrifuge tube and were diluted with isolation buffer containing 20 mM HEPES pH 7.3, 150 mM KCl, 10 mM MgCl₂, 2 mM DTT, 100 μg/ml CHX (at least one in three for the monosome fractions and one in five for the polysome fractions). The diluted fractions were put on top of 1 M sucrose cushion (made in isolation buffer) and centrifuged at 180,000×g (SW32Ti rotor) at 4 °C overnight. The pellets were rinsed gently with isolation buffer, then incubated with 50 μl of ribosome isolation buffer, and stored on ice for 1 h to allow the resuspension of the isolated ribosomes. Isolated ribosomes (50 nM final concentration) were incubated with recombinant HTT exon 1 proteins (500 nM final concentration) in an isolation buffer for 10 min at room temperature. The samples were loaded on top of 10–50% sucrose gradients and centrifuged at 280,000×g (SW41Ti rotor) at 4 °C for 2 h. The fractions containing monosomes were collected. Protein was precipitated and used to run western blotting assays. GST-HTT protein was produced in bacteria as described before[139].

**In vitro translation assay.** Recombinant human HTT proteins were purchased from Coriell life sciences (HTT-Q23, 1-3144, TEV, FLAG C-TE, # CH02228, HTT-Q48, 1-3144,TEV, FLAG C-TE, #CH02230). In vitro translation assays (IVTs) were performed using Flexi® Rabbit Reticulocyte Lysate System (Promega #L4540) following the manufacturer's recommendations. Briefly, firefly luciferase mRNA (included in the kit) was used to measure translational regulation by HTT proteins

comparing to BSA (as the control). Recombinant HTT proteins (1 mg/ml) or BSA (1 mg/ml) in 10 mM Tris pH 8.0, 1 mM EDTA, were further dissolved in Tris buffer (50 mM Tris-HCl pH 7.4, 500 mM NaCl, 10% glycerol, 0.1% CHAPS, and 1 mM EDTA). In total, 1 μl of Q23 or Q48 or BSA containing desired concentrations were added to 25 μl or 50 μl rabbit reticulocyte IVT containing luciferase mRNA for 90 min at 30 °C (IVT mixture). In control reactions, Q48 was added after 90 min of IVT for 5 min. In all, 2 μl of IVT mixture was added to 25 μl of luciferase assay reagent, and luminescence activities were measured in each well every 30 s over a period of 1 h with 500-ms integration time (data is presented for every 5 min) using a FlexStation3 plate reader (Molecular Devices). We avoided repeated freezing and thawing of recombinant HTT proteins as it drastically affected the activity in the IVT assay.

**Ribosome run-off assay.** Mouse striatal cells were plated at 60–70% confluency, on the next day were incubated with vehicle (DMSO) or harringtonine (2 μg/ml final concentration) for indicated timepoints or puromycin (100 μg/ml) for 20 min at 37 °C. The cells were immediately incubated with CHX (100 μg/ml) for 10 min and then scraped. Polysome profiles for each sample were collected and area under the curve for PS and 80S (MS) peaks in control and HD-homo cells, using Peak-Chart (v. 2.08, BRANDEL), and expressed as a ratio of PS/MS (Supplementary Fig. S1).

**Postmortem HD brain samples.** Postmortem frozen human brain tissue (Caudate nucleus) samples of HD-affected patients and normal donor controls used (Supplementary Table 2) in this study were obtained from Human Brain and Spinal Fluid Resource Center, VA West Los Angeles Healthcare Center, 11301 Wilshire Blvd. Los Angeles, CA 90073, which is supported in part by the National Institutes of Health (NIH) Neurobiobank (HHSN-271-201300029C) and the US Department of Veterans Affairs with informed consent from the donors. Human tissue collected from the NIH NeuroBioBank was overseen by institutional review board PCC #: 2015–060672, VA Project #: 0002 and were analyzed under ethical and safety guidelines approved by the Scripps Research Institute and its Institutional Review Board.

**Western blot analysis.** The cells were lysed in the lysis buffer containing 20 mM HEPES pH 7.3, 150 mM KCl, 10 mM MgCl$_2$, 2 mM DTT, 100 μg/ml CHX, 0.5% v/v Triton X-100, 20 U/ml RNasin, and EDTA-free protease inhibitor cocktail (Roche) and an RNA concentration A260 reading of 10 OD, loaded on a 30–50% sucrose gradient. Individual fractions (250 μl) were collected, the protein was precipitated using the methanol/chloroform method, and loaded for western blots analysis using antibodies to detect indicated endogenous protein. To examine Htt interaction with ribosomes HD-het cells were treated with vehicle (HEPES buffer) or puromycin (100 μg/ml) for 30 min at 37 °C followed by lysis and sucrose gradient fractionation. Human tissue was homogenized in binding/lysis buffer (50 mM Tris (pH 7.4), 150 mM NaCl, 10% glycerol, and 1.0% Triton X-100) with protease and phosphatase inhibitors, followed by a brief sonication for 6 s at 20% amplitude. Total proteins were precipitated from monosome/polysome sucrose fractions using methanol/chloroform. Protein pellets were resuspended in a buffer containing 10 mM Tris-HCl pH 8 and 0.1% SDS and used to run western blotting assays. The lysates or fractions were subjected to western blotting, as described previously[103,155].

**Ribosome profiling.** Global RNase foot-printings were performed during three independent rounds of cell cultures ($n = 3$). For each round of global footprinting, mouse immortalized striatal cells (i.e., control, HD-het, and HD-homo cells) were plated in 15-cm dishes at a confluency of 70%. The following day, the medium was changed, and after 2 h, the cells were incubated with CHX (100 μg/ml) for 10 min as in previous studies[95,96]. Cells were then scraped and washed with cold PBS (containing 100 μg/ml CHX) twice. During the second wash, 5% of cells were transferred to different tubes and were lyzed by adding 700 μl of QIAzol lysis reagent. Total RNAs of these samples were isolated using miRNeasy Mini Kit (Qiagen) for mRNA sequencing. After the second wash, the rest of the cells were lysed in a lysis buffer containing 20 mM HEPES pH 7.3, 150 mM KCl, 10 mM MgCl$_2$, 2 mM DTT, 100 μg/ml CHX, 0.5% v/v Triton X-100, 20 U/ml RNasin, and EDTA-free protease inhibitor cocktail (Roche). The cell lysates were passed 20 times through a 26-G needle and incubated on ice for 15 min, then centrifuged at 21,000×g for 15 min. Supernatants were transferred to different tubes. Equal total RNA amount of each sample was used for global RNase footprinting as follows; for each A260 absorbance unit of the lysates 60 units of RNaseT1 (Thermo Fisher Scientific) and 0.6 μl of RNaseA (Ambion) were added and the samples were incubated at 25 °C for 30 min. RNase-treated samples were immediately loaded on 10–50% sucrose gradients and centrifuged at 280,000×g (SW41Ti rotor) at 4 °C for 2 h. Gradients were fractionated using a gradient fractionator and UA-6 detector, 254-nm filter (ISCO/BRANDEL). Fractions containing 80S peaks of each sample were collected, and their RNAs were isolated using a miRNeasy Mini Kit (Qiagen). The area under the curve for PS and 80S (MS) peaks in control and HD cells, using PeakChart (v. 2.08, BRANDEL), and expressed as a ratio of PS:MS.

**Generation of cDNA libraries from ribosome-protected mRNAs.** The following procedure was performed for all the RNA samples simultaneously. In total, 20 μg of each sample was run on a 15% TBE-Urea gel (Novex) along with 26 and 32 nt RNA markers. The gel containing each sample was excised between two markers. RNAs were extracted from gel pieces by incubating gel slurries with nuclease-free water overnight at 4 °C and precipitated using RNase-free isopropanol and then eluted in nuclease-free water. T4 polynucleotide kinase (NEB) was used to catalyze the addition of 5′ monophosphate and removal of the 3′ phosphate in the RNA fragments to leave a 3′ hydroxyl terminal needed for adapter ligation. RNA was purified using the Zymo clean and conc-5 kit (Zymo Research, Cat. no. R1013). Ribosomal RNA was depleted from the samples using TruSeq total RNA rRNA-depletion protocol (Illumina, Cat. no. RS-122-2201) and then RNA samples were purified using Agencourt RNAClean XP beads (Beckman Coulter).

**Generation of cDNA libraries and sequencing.** NEXTflex small RNA-Seq Kit v3 (Perkin Elmer) was used to ligate 5′ and 3′ adapters to purified RPF fragments, which then were reverse-transcribed and amplified (14 cycles) to generate cDNA libraries. Libraries were cleaned up using NEXTflex Cleanup beads, pooled and sequenced in the NextSeq 500 (V2) using single-end 50 bp chemistry at the Scripps Genomic Core, at FL, USA.

**Generation of mRNA-Seq libraries.** Total RNA extracted from the cultured striatal cells as noted under Ribosome profiling were used for mRNA-seq library preparation. NEBNext Ultra II Directional kit (NEB, Cat. no. E776) with the NEBNext poly(A) mRNA magnetic isolation module (NEB, Cat. no. E7490) was used to generate mRNA-Seq libraries. Briefly, 400 ng of high-quality total RNA was used to purify poly(A) mRNA, fragmented, reverse-transcribed with random primers, adapter-ligated, and amplified according to the manufacturer's recommendations. The final libraries were validated on the bioanalyzer, pooled, and sequenced on the NextSeq 500 using paired-end 40 bp chemistry.

**Ribo-Seq, RNA-Seq quality control, and mapping the reads to UCSC browser.** RNAseq reads were trimmed using Cutadapt v1.18[156] with the following parameters: -a AGATCGGAAGAGCACACGTCTGAACTCCAGTCA -A AGATC GGAAGAGCGTCGTGTAGGGAAAGAGTGT–minimum-length=15 –pair-filter=any. For Ribo-Seq reads, 3′ adapters were trimmed using Cutadapt with the following parameters: -a TGGAATTCTCGGGTGCCAAGG–minimum-length 23. The reads were further trimmed using Cutadapt to remove four bases from either side of each read accordingly to the NEXTflex™ Small RNA Trimming Instructions (cutadapt -u 4 -u -4). Fastq files were checked for quality control with FastQC v0.11.8. Both RNA-Seq and Ribo-Seq reads were next mapped to a library of mouse rRNA and tRNA sequences using Bowtie v1.1.2. Any reads mapping to these abundant contaminants were filtered out. The remaining reads were then aligned to the mouse transcriptome with RSEM v1.3.0[157] using the GRCm38.p5 genome annotation and the comprehensive gene annotation from Gencode (M16 release) as transcriptome reference. Reads with a mapping quality <5 were discarded. Cleaned bam files were converted to bigWig files with Bedtools v2.27.0[158] for visualization using the UCSC Genome Browser. For the Euclidian distance analyses, gene expression was quantified with RSEM v1.3.0, and comparison plots were generated in R using DESeq2 v1.22.2[159] and ggplot2 v3.3.0 packages. Statistical testing was done using DESeq2 with a two-tailed Wald test and adjusted for multiple comparisons using the procedure of Benjamini–Hochberg[160].

**Ribosome occupancy (Anato2Seq) analysis.** The raw Ribo-Seq reads were clipped of adapter sequence (TGGAATTCTCGGGTGCCAAGG) using Cutadapt (version 1.18)[156] with the following command: cutadapt -f fastq -a CTGTAGGCACCATCAAT–minimum-length=23 <input > .fastq -o  . fastq. A 4 bp secondary trim from either end of the reads was performed also using Cutadapt. Mouse rRNA sequences were retrieved from NCBI[161] with the following accessions: NR_003279, NR_003278, NR_003280, NR_030686. Ribo-Seq and RNA-Seq reads aligning to these sequences were removed using bowtie (version 1.0.1)[162] with the following command: bowtie -v 3–norc <path_to_rRNA_indices > -q <input > .fastq –un  .fastq. The remaining reads were then mapped using bowtie to the RefSeq[163] mouse transcriptome downloaded from ftp://ftp.ncbi.nlm.nih.gov/refseq/M_musculus/mRNA_Prot/ on October 9, 2018. The following command was used: bowtie -a -m 100 -l 25 -n 2 -S–norc <path_to_transcriptome_indices > -q <input > .fastq  .sam.

The sam alignment files were parsed with an in-house python script to count the number of reads aligned to each gene using an exon union approach[164]. The Ribo-Seq reads were assigned to mRNA coordinates using an offset of 14 nucleotides downstream of the 5′ end of the reads. The counts included uniquely mapped reads i.e., reads that mapped to one location only in the mouse transcriptome plus reads that mapped up to 3 locations in the transcriptome which were weighted by the number of their mapped locations (up to 3).

For differential expression analysis using Anota2seq[97], the number of reads aligning to annotated CDS regions was used for Ribo-Seq while for RNA-Seq the number of reads aligning to the entire transcript was used. These counts were input to anota2seq (version 1.5.2) with the parameters dataType = "RNAseq", filterZeroGenes = FALSE, normalize = TRUE, transformation = "TMM-log2". All

statistical tests within the anota2seq package are two-tailed. Violin plots were generated using R version 4.0.2 (2020-06-22) to show the distribution of ribosome occupancy changes[165].

**Ribosome pause (PausePred) analysis**. The command-line version of the PausePred software[100,164] was run with the following parameters for each replicate (control, HD-het and HD-homo): fold change: 5; window size: 1000 nucleotides; read lengths: 26–32 nucleotides; window coverage: 5. Individual offset values were assigned according to metagene analysis for each read length (26–32 nucleotides) accounting for mismatches at the 5′ ends of the reads. For genes with detected pauses, the center of ribosome density[166] was determined using an in-house python script.

**cDNA preparation and real-time PCR**. A sucrose density gradient centrifugation was carried out using control and HD-homo cells, and ribosome fractions were collected. RNA was extracted from the fractionated samples following lysis in Trizol reagent. In total, 250 ng of RNA was used to prepare cDNA using Takara primescript™ kit (Cat no. 6110A) using random hexamers. The qRT-PCR of genes was performed with SYBR green (Takara RR420A) reagents. Primers for all the genes were designed based on sequences available from the Harvard qPCR primer bank.

The total mRNA-Seq data of control and HD-homo cells was used to estimate the Fmr1 and actin mRNA reads. The mRNA was isolated from the striatal tissue of unaffected and HD patient striatum (grade 1 and grade 2) and qPCR for FMR1 or GAPDH were done as described[103]. Relative mRNA expression of Fmr1 was determined after normalization with Actin or Gapdh transcripts. The list of PCR primers used in this study is listed in Supplementary Table 3.

**Statistical analysis**. Data are presented as mean ± SEM as indicated. Except where stated all experiments were performed at least in three biological replicates and repeated at least twice. Statistical comparison was performed between groups using two-tailed Student's $t$ test, one-way analysis of variance (ANOVA) followed by Tukey's multiple comparison test or Bonferroni's multiple comparisons test and two-way ANOVA or two-way repeated measures ANOVA followed by Tukey's multiple comparison test or Bonferroni post-hoc test as indicated in the figure legends. Significance was set at $P < 0.05$. All statistical tests were performed using Prism 7.0 (GraphPad software).

**Reporting summary**. Further information on research design is available in the Nature Research Reporting Summary linked to this article.

## Data availability

The complete dataset from the analysis of the HTT interactors from healthy and HD fibroblasts (raw files, identification data, and data analysis files) can be obtained via ProteomeXchange with identifier PXD017115. The data for the Ribo-Seq and RNA-Seq reported in this study are openly available in Gene Expression Omnibus at accession number GSE146675. UCSC browser information to view genome browser hub with the RNA-Seq and Ribo-Seq data: To find genes of your interests, go to the Genome browser website (http://genome.ucsc.edu) and then click on "My Data > Track Hubs". Then paste the link https://data.cyverse.org/dav-anon/iplant/home/rmi2lab/Hub_Collaborations/Srini/hub.txt in the "url" field and click on "Add hub." After the hub is loaded, go to "Genomes > Mouse GRCm38/mm10". Once you reach the actual browser window, you will have to scroll down to the bottom menu. Find a section (the one at the top of the menu) named "Srini" to activate the different tracks. The data supporting the findings of this study are available from the corresponding authors upon reasonable request. Source data are provided with this paper.

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

## Acknowledgements

We would like to thank Melissa Benilous for administrative help, and members of the lab for continuous support and collaborative atmosphere. We like to thank members at the Scripps proteomics and genomic core for their help and expertise. This research was partially supported by a training grant in Alzheimer's drug discovery from the Lottie French Lewis Fund of the Community Foundation for Palm Beach and Martin Counties; grant awards from NIH/NINDS R01-NS087019-01A1, NIH/NINDS R01-NS094577-01A1, and a grant from Cure for Huntington Disease Research Initiative (CHDI) foundation.

## Author contributions

S.S made the initial observations and further conceptualized the project and revised the paper together with N.S., M.S., U.N.R.-J., and A.M. M.E. carried out polysome profiles, and related biochemical experiments, prepared volcano plots, and pause density graphs. P.P.K. generated cDNA library. E.R. and J.B. carried out bioinformatics and Ribo-Seq/mRNA-Seq overlap tracks on the UCSC Genome Browser using track hubs. A.M. generated the data for the triplet periodicity, metagene, pause analysis, the center of ribosome density, and differential gene expression analysis. N.S. carried out IP, additional polysome-related biochemical experiments, and analysis of the data. N.G. helped in Western blotting experiments. N.T.U. carried out STED imaging and colocalization analysis. M.S., U.N.R.-J., K.F., and J.F. supported in qPCR, profiling, and western blotting. S.S. analyzed the data and wrote the paper with input from co-authors.

## Competing interests

The authors declare no competing interests.
