## [Peer Review File · Nature Communications]

Reviewers' comments:

Reviewer #1 (Remarks to the Author):

Review for the manuscript NCOMMS-20-05400 by Dr. Subramaniam and co-authors entitled "Mutant huntingtin stalls ribosomes and represses protein synthesis independent of Fragile X mental retardation protein"

Mutations in mRNA-binding proteins (RBPs) such as SMN, FUS, Ataxin2, and Fmr1 promote neurodegenerative diseases. It is mostly unknown how physiological signals directly regulate ribosome stalling and how their dysregulation affects neurodegenerative disease processes. Huntington disease (HD) is a neurodegenerative disorder caused by the expansion of a polyglutamine tract in the huntingtin (mHtt) protein. mHtt aberrantly alters cellular and molecular functions, including vesicle trafficking, nuclear transport defects, and transcription, but its role in regulating protein synthesis is not well understood.

In this study, the authors tried to address how protein synthesis is suppressed in HD cells. They claimed that monosome(MS)/polysome(PS) ratio in the presence of harringtonine (2 min), called a harringtonine-based ribosome run-off assay (RRA), is an indicator for translation repression. In combination with puromycin-incorporation, they claimed that the depletion of mHtt enhances protein synthesis and increases the speed of ribosome translocation. The co-IP results suggested the interaction of mHtt with the ribosomes, and mHtt inhibits protein synthesis with the in vitro translation assay. The inhibitory effect of mHtt in translation is independent of Fmr1. Based on these results, they propose that mutant huntingtin stalls ribosomes and represses protein synthesis independent of Fragile X mental retardation protein.

The proposal is potentially interesting. However, the quality of results is not sufficient enough to support their proposal that mHtt promotes ribosome stalling and inhibits protein synthesis. There is no direct evidence to demonstrate that mHtt inhibits translation elongation. It is also unclear how mHtt interacts with translating ribosomes to inhibit translation elongation step(s). More experiments need to be done to strengthen their proposal.

Major points

1. Figs. 3, 4, 5, 6 in the pdf are miss-assigned and should be Figs. 4, 5, 6, and 3. Fig. 2F is missing in Figure 2.
2. Fig. 1G: In this study, the authors claimed that the monosome(MS)/polysome(PS) ratio in the presence of harringtonine (2 min) that is called as a harringtonine-based ribosome run-off assay (RRA), is an indicator for translation repression by wtHtt or mHtt. Delay of the polysome depletion after harringtonine treatment is consistent with their model. However, the other possibility is that translation initiation in HD-hetero (STHdhQ7/Q111) and HD-homo (STHdhQ111/Q111) is resistance to harringtonine treatment. The authors should exclude this possibility.
3. Fig. 1G: The authors also claimed that the ribosomes appeared to complete their translation in both the HD-homo and the control cells within five minutes of the harringtonine treatment. However, the significant amount of polysomes remained in both cells five minutes after harringtonine treatment. They should describe the content of the remaining polysome five minutes after harringtonine treatment. Besides, polysome analysis after more prolonged treatment than five minutes should be demonstrated.
4. Fig. 2D-E: It is not very easy to evaluate the experiments shown in Fig. 2 because of Fig. 2F is missing. For more precise evaluation of the function of wtHtt and mtHtt in the repression of translation elongation, the amount of polysome in WT, Htt-KD, HD-hetero(STHdhQ7/Q111), HD-homo (STHdhQ111/Q111) striatal cells should be demonstrated in the same panel. Given that mHtt (111Q) is aggregate-prone, the authors should explain how aggregate-prone mHtt (111Q) represses translation elongation more efficiently than wtHtt (8Q).

5. Fig. 6F in the text (Fig. 3F in the merged pdf): Both wtHtt and mHtt were detected in the 40S, 60S, and 80S (MS) and polysome fractions in the sucrose gradients in control and HD-het striatal cells with Western blot analysis. However, ribosome-association seems to be very weak. In contrast to the statement that wtHtt (8Q) represses translation in cells (Fig. 2), wtHtt was not co-purified. It is unclear whether the interaction is sufficient enough for inhibition of translating ribosomes. As the putative role of HEAT repeats was discussed, it is possible to examine its role in the interaction with ribosomes.

6. Fig. 3 in the text (Fig. 4 in the merged pdf): The authors demonstrated that mHtt inhibits protein synthesis in vitro with the recombinant human FL-HTT (23Q) as wtHTT and FL-HTT (48Q) as mHTT. For consistent arguments with the results of RRA with FL-HTT (Q7) and FL-HTT (Q111), the in vitro experiments with FL-HTT (Q7) and FL-HTT (Q111) should be performed. In the same context, for consistent arguments with the results of co-purification with mHTT (23Q) and mHTT(48Q), the RRA should be performed with HD-hetero (STHdhQ23/23), HD-hetero (STHdhQ23/48) and HD-homo (STHdhQ48/Q48).

7. Fig. 8: The identification of the Htt targets is crucial to support their proposal. The putative Htt-targets identified in the HD-homo cells and the HD-het cells should be validated with western blot analysis nor in vitro translation assay with the target mRNAs. In Fig. 8C, the average relative positions of paused codons in HD-control should also be demonstrated with that in HD-homo and HD-het cells.

Minor points

1. The stoichiometry between wHtt and mHtt proteins and translating ribosomes is vital information to discuss the mechanism of translation repression by these proteins.
2. F3A: Is the total ribosome amount in HD-Het higher than control?
3. F6C: Control is missing in ribosome pause density.

Reviewer #2 (Remarks to the Author):

Review of Nature Communication # NCOMMS-20-05400

Mutant huntingtin stalls ribosomes and represses protein synthesis independent of fragile X mental retardation protein

Mechanism of Huntington disease has been studied for the past two decades since the gene mutation has been localized to the gene huntingtin (Htt). A number of affected pathways have been well described, which include mutant Htt (mHtt) affects vesicle trafficking, causes nuclear transport defects and abnormal gene transcription.

In the current study, the authors have shown a new function of wild type HTT in inhibiting protein synthesis. More importantly, they have found a new HD mechanism, which is that mHTT can inhibit protein synthesis to affect the cell function through aberrantly binding to the ribosomes during elongation. Although the majority studies were done with mouse striatal immortalized cell lines derived from HD knock in mouse models, they also used fibroblast cell lines from patients with HD and further confirmed their finding using brain tissues of patients with HD. The authors have used several advanced techniques and produced large quantities of beautiful results to support their conclusions.

Only a few minor concerns that authors need to make changes to improve the quality of the manuscript.

The majority of the work were done in the HD cell lines, it will be more accurate to include HD cell lines in the paper's title. Such as:

Mutant huntingtin stalls ribosomes and represses protein synthesis in HD knock in striatal cell lines.

Fig 8B Ribosome labeling seems confusing. The legend is black, but the graph is blue.

Figure 6 has no number

Fig S6 It will be helpful to have a graph summarize all the findings in the experiment with statistics.

Reviewer #3 (Remarks to the Author):

This study presents a novel mechanistic defect of mHtt (polyQ expanded mutant Huntingtin) on global translation inhibition and ribosome stalling which could be exploited for therapeutics for Huntington Disease (HD). The authors first showed the mHtt protein promotes global translation inhibition and depleting mHtt can release this inhibition. Ribosome run-off assay also indicated a slower translocating ribosome in the HD-homo compare with the control cells and knock down mHtt can rescue the stalling. Then through a series of biochemical and imaging study, they demonstrated mHtt's inhibition effect on translation is by interfering with the translation machinery. Finally, they performed high-throughput mRNA-seq from the run-off assay and ribo-seq analysis to identify mRNAs of which the translation profile was changed in the polyQ containing cells.

Overall, the authors employed reasonable experimental design to prove the novel mechanism on the translation inhibition by ribosome pausing induced by mHtt protein. Most of the results are well presented, although some interpretations are somewhat confusing and need to be clarified. While compelling, several data are not consistent in different figures/assays and cannot fully support the conclusions, as listed below. These issues are essential to be addressed before publication consideration.

Major concerns

1. Figure 1A and 1B are not consistent. In 1A, the blue line is higher than the green line at the polyribosome part. but it didn't show this difference in the quantification in 1B. And if there are no difference between control and HD-het, why there is translation inhibition in HD-heter in 1C,D?
2. In Figure 1C, the western blotting of wtHtt/mHtt showed there was much less Htt in mouse HD-het and HD-homo striatal cells compared to control. Why is that? In the patient derived fibroblasts, there was not such differences, as shown in Figure 1E.
3. In Figure 2A-C, depletion of mHtt barely rescued the translation defects. The effect by mHtt is less significant than the one from wtHtt, even though mHtt was reduced more than wtHtt. This data cannot support the claim that "although both wtHtt and mHtt inhibit protein synthesis, mHtt inhibitory effect is much stronger than the wtHtt in cells".
4. The whole manuscript is largely based on the PS/MS ratios, but it is not clear how the ratio was calculated. Which part of area was measured as MS and which part was treated as PS? And how was the area measured? This need to be described in Method. And a representative diagram needs to be included in the figure. In particular, the labeling of polysome part was not consistent in different figures. Some started from the di-ribosome peak, some started from the tri-ribosome peak. If this is not precisely defined, it will bring variation to the data and might lead to wrong conclusions. This problem also needs to be addressed regarding the ribo-seq experiment and the ribosome runoff assay.
5. Many figures were messed up. The current Figure 4 should be Figure 3 in the text. Figure 5 should be Figure 4, Figure 6 (not labeled) should be Figure 5. And Figure 3 should be Figure 6. The expected figure # will be used in the comments.

6. In Figure 3, the mHtt did not have much more translation inhibition than wtHtt. It requires higher concentration/amount to show slight effects. This cannot explain the profound translation inhibition observed in vivo, as in Fig. 1 and 2. Particularly, mHtt level was actually much lower than wtHtt as shown in Figure 1C.
7. The dynamic data in Figure 3D seemed not a proper piece of evidence indicating the polyQ interfering with the translation, but rather a slower deterioration of the luciferase signal compared with the BSA control. Tracking luciferase activity by choosing several time points during the 90min IVT reaction could be a better experiment.
8. The data in Figure 5H is not consistent with Figure 2E. the basal level of PS/MS ratio was not changed in 2E. Similar in Figure 3A (actually Figure 6A), the ribosome profile (PS/MS) presented here showed difference between control and HD-het cells, however contradictory to the data shown in Figure 1B, in which the quantification suggests there is no significant difference.
9. Figure 6B, samples from all fractions need to be run on one gel in order to know the relative distribution and abundance of the proteins. The current way of data presentation missed a lot of information.
10. Figure 6C, why using puromycin instead of harringtonine, which was used in all the other data? In fractions 3-6, the reduction of FMRP and RPL7 was not obvious as claimed, and the signal was way too saturated to know the relative quantity.
11. In figure 8A, why HD-heter and HD-homo has opposite changes? According to all other data in the manuscript, HD-heter and HD-homo have similar trend of changes, with higher magnitude in HD-homo. It is hard to understand the rationale of opposite trend of relative positions of ribosomes on mRNA in these two cells. And more importantly, the patients are actually HD-heter. How to understand the relevance of this data to patients? The HD-heter data in Figure 8A would not support the proposed model. Better to perform polysome run-off mRNA-seq in patient cells as well.
12. The author showed the enrichment of the single paused codon in the polysome run-off mRNA-seq data (Figure 8D). The current manuscript will benefit more by validating and discussing more of the hits from the two sets of sequencing data, such as a detailed look into if any specific pathway or motif are enriched in those stalled mRNAs targets. The data is not accessible from the link the author provided (Line 318).

Minor points

1. There is no figure legend for Figure 3D.
2. Figure 5G, the ribosome profiling lines need to use more distinct colors.
3. In Line 246-247, "As a positive control, we detected Caprin1, a previously 247 known interactor of mHtt". Need to cite Figure 6F.
4. Figure 4C and Figure 8D the two volcano plots are not very clear. The genes and the corresponding dots need to be more clearly labeled.
5. The total Fmr1 RNA level should be measured in Figure 4E.
6. Several typos eg. figure 5E (actually Figure 4E) "wtHtt" instead of "mtHtt" and Figure 8D "targets".
7. Line 124, double "compared to"
8. Line 410, "interfere" instead of "interferes"

We thank the referee for the careful and insightful review of our manuscript. We address all of the concerns of the referee here indicated in blue.

Reviewers' comments:

Reviewer #1 (Remarks to the Author):

Review for the manuscript NCOMMS-20-05400 by Dr. Subramaniam and co-authors entitled "Mutant huntingtin stalls ribosomes and represses protein synthesis independent of Fragile X mental retardation protein"

Mutations in mRNA-binding proteins (RBPs) such as SMN, FUS, Ataxin2, and Fmr1 promote neurodegenerative diseases. It is mostly unknown how physiological signals directly regulate ribosome stalling and how their dysregulation affects neurodegenerative disease processes. Huntington disease (HD) is a neurodegenerative disorder caused by the expansion of a polyglutamine tract in the huntingtin (mHtt) protein. mHtt aberrantly alters cellular and molecular functions, including vesicle trafficking, nuclear transport defects, and transcription, but its role in regulating protein synthesis is not well understood.

In this study, the authors tried to address how protein synthesis is suppressed in HD cells. They claimed that monosome(MS)/polysome(PS) ratio in the presence of harringtonine (2 min), called a harringtonine-based ribosome run-off assay (RRA), is an indicator for translation repression. In combination with puromycin-incorporation, they claimed that the depletion of mHtt enhances protein synthesis and increases the speed of ribosome translocation. The co-IP results suggested the interaction of mHtt with the ribosomes, and mHtt inhibits protein synthesis with the in vitro translation assay. The inhibitory effect of mHtt in translation is independent of Fmr1. Based on these results, they propose that mutant huntingtin stalls ribosomes and represses protein synthesis independent of Fragile X mental retardation protein.

The proposal is potentially interesting. However, the quality of results is not sufficient enough to support their proposal that mHtt promotes ribosome stalling and inhibits protein synthesis. There is no direct evidence to demonstrate that mHtt inhibits translation elongation. It is also unclear how mHtt interacts with translating ribosomes to inhibit translation elongation step(s). More experiments need to be done to strengthen their proposal.

Reviewer # 1 indicates that our finding is potentially interesting; he/she indicates the following comments and recommendations.

Major points

1. Figs. 3, 4, 5, 6 in the pdf are miss-assigned and should be Figs. 4, 5, 6, and 3. Fig. 2F is missing in Figure 2.

We apologize for this error, thank the reviewer for the kindness in bringing this to our attention. We have now carefully assigned the figure numbers to match the results and

discussion in the revised manuscript.

2. Fig. 1G: In this study, the authors claimed that the monosome(MS)/polysome(PS) ratio in the presence of harringtonine (2 min) that is called as a harringtonine-based ribosome run-off assay (RRA), is an indicator for translation repression by wtHtt or mHtt. Delay of the polysome depletion after harringtonine treatment is consistent with their model. However, the other possibility is that translation initiation in HD-hetero (STHdhQ7/Q111) and HD-homo (STHdhQ111/Q111) is resistance to harringtonine treatment. The authors should exclude this possibility.

The reviewer raises a valid point. After harringtonine, the time taken for the polysome depletion is correlated with the speed of translation elongation (Arguello et al., 2018; Conn and Qian, 2013; Dana and Tuller, 2012; Eastman et al., 2018; Fresno et al., 1977; Ingolia et al., 2011; Lee et al., 2012; Shcherbakov et al., 2019). However, HD cells may be resistant to harringtonine, as pointed by the reviewer. To exclude this possibility, we compared the ribosome profiling of control and HD cells in the presence of puromycin, which is also used to assess the ribosome depletion from translating mRNA, indicative of ribosome run-off (Azzam and Algranati, 1973; Darnell et al., 2011; Sivan et al., 2007). Puromycin treatment showed an enhanced PS/MS ratio in HD cells, reflecting a much slower ribosome run-off than control cells. Thus, both harringtonine and puromycin experiments showed that the polysome depletion from the translating mRNA occurs much slower in HD cells than control cells. This new data is included in the revised manuscript (Fig. S5).

3. Fig. 1G: The authors also claimed that the ribosomes appeared to complete their translation in both the HD-homo and the control cells within five minutes of the harringtonine treatment. However, the significant amount of polysomes remained in both cells five minutes after harringtonine treatment. They should describe the content of the remaining polysome five minutes after harringtonine treatment. Besides, polysome analysis after more prolonged treatment than five minutes should be demonstrated.

The reviewer asks another valid point and recommends polysome analysis after 5 minutes of harringtonine. As suggested by the reviewer, we carried out ribosome profiling at 0 (basal), 2, 5, and 8 minutes after harringtonine treatment. In parallel, estimated protein synthesis to determine the remaining contents of the polysome. As shown in the revised Fig. 1 E-H (previously Fig. 1G), we found a high PS/MS ratio in HD cells compared to control at 0 and 2 minutes after harringtonine treatment, which gradually decreased and was not significantly different at 5 and 8 minutes (Fig. 1E, F) by two-way ANOVA, Tukey's multiple comparisons tests. Interestingly, the protein synthesis (as measured by puromycin incorporation) is significantly lower in HD cells at 0 (basal) and 2 minutes of harringtonine treatment compared to control cells (Fig. 1G, H). More than 80% of ribosomes completed mRNA translation by 8 minutes of harringtonine. Thus, despite enhanced ribosome occupancy in HD cells, the mRNA

translation seems to occur at a slow pace, suggesting a potential elongation deficit in HD.

4. Fig. 2D-E: It is not very easy to evaluate the experiments shown in Fig. 2 because of Fig. 2F is missing. For more precise evaluation of the function of wtHtt and mHtt in the repression of translation elongation, the amount of polysome in WT, Htt-KD, HD-hetero(STHdhQ7/Q111), HD-homo (STHdhQ111/Q111) striatal cells should be demonstrated in the same panel. Given that mHtt (111Q) is aggregate-prone, the authors should explain how aggregate-prone mHtt (111Q) represses translation elongation more efficiently than wtHtt (8Q).

We apologize for the missing Fig. 2F and thank the reviewer for this important recommendation. We have included polysomes of wtHtt-KD and mHtt-KD in their respective panel (Fig. 2D, G). We also carried out additional experiments for Fig. 2F (now Fig. 2J, K), which demonstrates that polysome run-off much slower in mHtt-depleted cells compared to Htt-depleted cells.

The reviewer asks us to discuss aggregated mHtt role in repressing translation. We have dedicated a paragraph incorporating previous work and discussing how poly-Q expansion and its aggregates might suppress elongation (Line 468).

5. Fig. 6F in the text (Fig. 3F in the merged pdf): Both wtHtt and mHtt were detected in the 40S, 60S, and 80S (MS) and polysome fractions in the sucrose gradients in control and HD-het striatal cells with Western blot analysis. However, ribosome-association seems to be very weak. In contrast to the statement that wtHtt (8Q) represses translation in cells (Fig. 2), wtHtt was not co-purified. It is unclear whether the interaction is sufficient enough for inhibition of translating ribosomes. As the putative role of HEAT repeats was discussed, it is possible to examine its role in the interaction with ribosomes.

We rerun the polysome fractions treated with vehicle or harringtonine in the same gel (Fig. 6A). Both wtHtt and mHtt are found in the PS fractions. Upon harringtonine treatment, wtHtt and mHtt resediment to the lower fractions, indicating that the HTT and mHTT associate with elongating ribosomes. In a reconstitution (in vitro binding experiment), as suggested by the reviewer, we observed that mHTT binds more strongly to the ribosomal fractions than wtHTT. Although the reasons are unclear, HTT is modified by various PTMs, including SUMOylation, as we demonstrated before (Ehrnhoefer et al., 2011; Subramaniam et al., 2009). Because we purified these proteins using a bacterial expression system, which lacks efficient post-translational modifications (like glycosylation, phosphorylation, SUMOylation), we predict that some unknown PTM and additional signals may influence HTT interaction with ribosomes. So, we carried our HTT interaction studies in amino acids stimulation condition by IP-LC-MS/MS in human HD fibroblasts and found strong interaction of HTT and mHTT to the ribosomal proteins (Fig. 6F) as well as enrichment of the biological process and protein components related to the translation and ribosome/RNA binding components (Fig. S7). This data is now included in the revised text (line 268) and Fig. 6F, and Fig. S7.

We thank the reviewer for his/her thoughtful suggestion about examining HTT HEAT repeat domain interactions with the ribosomes. There are ~30 HEAT domains that span across the HTT proteins. We will systematically explore HEAT domain protein interactions with ribosomes and their roles in ribosome movement and protein synthesis in future studies.

6. Fig. 3 in the text (Fig. 4 in the merged pdf): The authors demonstrated that mHtt inhibits protein synthesis in vitro with the recombinant human FL-HTT (23Q) as wtHTT and FL-HTT (48Q) as mHTT. For consistent arguments with the results of RRA with FL-HTT (Q7) and FL-HTT (Q111), the in vitro experiments with FL-HTT (Q7) and FL-HTT (Q111) should be performed. In the same context, for consistent arguments with the results of co-purification with mHTT (23Q) and mHTT(48Q), the RRA should be performed with HD-hetero (STHdhQ23/23), HD-hetero (STHdhQ23/48) and HD-homo (STHdhQ48/Q48).

The reviewer asks us to carry out the RRA experiment using cell types (STHdhQ23/23), HD-hetero (STHdhQ23/48), and HD-homo (STHdhQ48/Q48). This is a great suggestion, but unfortunately, such knock-in cell lines are not available.

7. Fig. 8: The identification of the Htt targets is crucial to support their proposal. The putative Htt-targets identified in the HD-homo cells and the HD-het cells should be validated with western blot analysis nor in vitro translation assay with the target mRNAs. In Fig. 8C, the average relative positions of paused codons in HD-control should also be demonstrated with that in HD-homo and HD-het cells.

We thank the reviewer for this important suggestion. We have validated five targets.

For example, Mfsd10, an ion transporter, showed a single codon pause at position 1673 (GAG) and enhanced RPF/mRNA reads in the HD-homo and HD-Het cells (Fig. 9A, arrow). Mgp, N-methylpurine-DNA glycosylase, showed a single codon pause at 100 (AGC) and high RPF reads in HD-het (seventeen-fold), and HD-homo (six-fold) compared to control cells, with RPF tilted more toward 3' in the HD-het cells (Fig. 9D, arrow). But these targets showed an overall reduction in the protein levels in HD cells compared to control (Fig. 9G), indicating these targets are controlled by ribosome stalling and posttranslational regulations.

We show HD exclusive pauses compared to control to generate Fig 8C. This information is now included in Fig. 8C.

Minor points

1. The stoichiometry between wHtt and mHtt proteins and translating ribosomes is vital information to discuss the mechanism of translation repression by these proteins.

We thank the reviewer for this important suggestion. We have discussed how stoichiometry of wHtt and mHtt may impact the mRNA translation (line 468).

2. F3A: Is the total ribosome amount in HD-Het higher than control?

This was representative data, which is now revised. We do not observe any significant differences in the ribosome amount or proteins between control and HD (e.g., Fig. 1A, 6A, 6D).

3. F6C: Control is missing in ribosome pause density.

Ribosome pause density is calculated in compared to control; this information is now included in the revised Fig 8C (previously Fig 6C).

Reviewer #2 (Remarks to the Author):

Review of Nature Communication # NCOMMS-20-05400

Mutant huntingtin stalls ribosomes and represses protein synthesis independent of fragile X mental retardation protein

Mechanism of Huntington disease has been studied for the past two decades since the gene mutation has been localized to the gene huntingtin (Htt). A number of affected pathways have been well described, which include mutant Htt (mHtt) affects vesicle trafficking, causes nuclear transport defects and abnormal gene transcription.

In the current study, the authors have shown a new function of wild type HTT in inhibiting protein synthesis. More importantly, they have found a new HD mechanism, which is that mHTT can inhibit protein synthesis to affect the cell function through aberrantly binding to the ribosomes during elongation. Although the majority studies were done with mouse striatal immortalized cell lines derived from HD knock in mouse models, they also used fibroblast cell lines from patients with HD and further confirmed their finding using brain tissues of patients with HD. The authors have used several advanced techniques and produced large quantities of beautiful results to support their conclusions.

Only a few minor concerns that authors need to make changes to improve the quality of the manuscript.

The majority of the work were done in the HD cell lines, it will be more accurate to include HD cell lines in the paper's title. Such as:

Mutant huntingtin stalls ribosomes and represses protein synthesis in HD knock in striatal cell lines.

We thank the reviewer for indicating that our work is impressive and data supports our conclusions.

Based on the reviewer suggestions, we have modified the title as

“Mutant Huntingtin Stalls Ribosomes and Represses Protein Synthesis in a Cellular Model of Huntington disease.”

Fig 8B Ribosome labeling seems confusing. The legend is black, but the graph is blue. We thank the reviewer for pointing out the error. We now changed this in the Fig. 8B.

Figure 6 has no number

We have fixed this error.

Fig S6 It will be helpful to have a graph summarize all the findings in the experiment with statistics.

We have included an additional figure associated with Fig S6 (now Fig. S9), Fig. S12, with Spearman correlations.

Reviewer #3 (Remarks to the Author):

This study presents a novel mechanistic defect of mHtt (polyQ expanded mutant Huntingtin) on global translation inhibition and ribosome stalling which could be exploited for therapeutics for Huntington Disease (HD). The authors first showed the mHtt protein promotes global translation inhibition and depleting mHtt can release this inhibition. Ribosome run-off assay also indicated a slower translocating ribosome in the HD-homo compare with the control cells and knock down mHtt can rescue the stalling. Then through a series of biochemical and imaging study, they demonstrated mHtt's inhibition effect on translation is by interfering with the translation machinery. Finally, they performed high-throughput mRNA-seq from the run-off assay and ribo-seq analysis to identify mRNAs of which the translation profile was changed in the polyQ containing cells.

Overall, the authors employed reasonable experimental design to prove the novel mechanism on the translation inhibition by ribosome pausing induced by mHtt protein. Most of the results are well presented, although some interpretations are somewhat confusing and need to be clarified. While compelling, several data are not consistent in different figures/assays and cannot fully support the conclusions, as listed below. These issues are essential to be addressed before publication consideration.

Major concerns

1. Figure 1A and 1B are not consistent. In 1A, the blue line is higher than the green line at the polyribosome part. but it didn't show this difference in the quantification in 1B. And if there are no difference between control and HD-het, why there is translation inhibition in HD-heter in 1C,D?

We have revised this figure and show a representative image (Fig. 1A). Although some HD-het profiles (n = 8) appear to display enhanced polysome, they were not statistically significant compared to controls (n = 20) (Fig. 1B).

The reviewer asks an important question. He/she wonders why even though we are unable to see a significant difference at the polysome profile, we observed decreased protein synthesis. Although the reasons are unclear, we predict that in HD-het, the high ribosome occupancy is so subtle that we could not capture differences using ribosome profiling. However, measurement of protein synthesis via SUNSET assay, which is highly sensitive, was able to identify the subtle decline in protein synthesis in HD-het compared to control (Fig. 2C, D). In HD-homo, on the other hand, due to two copies of mHtt, the ribosome occupancy is drastically enhanced, resulting in a significantly increased PS/MS ratio and diminished protein synthesis.

2. In Figure 1C, the western blotting of wtHtt/mHtt showed there was much less Htt in mouse HD-het and HD-homo striatal cells compared to control. Why is that? In the patient derived fibroblasts, there was not such differences, as shown in Figure 1E.

The reviewer asks a valid question. HD-het and HD-homo consist of many CAG repeat coding for 111 glutamine (Fig. 1C), whereas patient-derived cells consist of 69 repeats (Fig. S3). Longer the CAG repeats, the expression seems to diminish either due to increased aggregates, decreased solubility, or diminished synthesis due to ribosome stalling (unpublished observations). Also, mHTT levels and aggregates status is determined in a cell-type dependent manner (Cisbani, 2012).

3. In Figure 2A-C, depletion of mHtt barely rescued the translation defects. The effect by mHtt is less significant than the one from wtHtt, even though mHtt was reduced more than wtHtt. This data cannot support the claim that “although both wtHtt and mHtt inhibit protein synthesis, mHtt inhibitory effect is much stronger than the wtHtt in cells”.

We agree with the reviewer that mHtt depletion did not completely rescue the translation defects. Despite a substantial reduction in mHtt depletion, the protein synthesis or PS/MS ratio remains inhibited in mHTT depleted cells, compared to wtHtt depleted cells. Thus we posit that the mHtt inhibitory effect is much stronger than wtHtt. However, we realize this conclusion is the amount of confusion as fold change in the protein synthesis is similar between wtHtt depleted, and mHtt depleted cells. Therefore, in the revised text, we have concluded “that both wtHtt and mHtt inhibit protein synthesis. (line 149)”

4. The whole manuscript is largely based on the PS/MS ratios, but it is not clear how the ratio was calculated. Which part of area was measured as MS and which part was treated as PS? And how was the area measured? This need to be described in Method. And a representative diagram needs to be included in the figure. In particular, the labeling of polysome part was not consistent in different figures. Some started from the di-ribosome peak, some started from the tri-ribosome peak. If this is not precisely

defined, it will bring variation to the data and might lead to wrong conclusions. This problem also needs to be addressed regarding the ribo-seq experiment and the ribosome runoff assay.

The reviewer raises an important point. We have provided a representative image of the area under the curve (Fig. S1) and indicated what area and how PS/MS ratio is measured in the revised method section, including Ribo-seq and the ribosome runoff assay, as recommended by the reviewer (line 636).

5. Many figures were messed up. The current Figure 4 should be Figure 3 in the text. Figure 5 should be Figure 4, Figure 6 (not labeled) should be Figure 5. And Figure 3 should be Figure 6. The expected figure # will be used in the comments.

We have resolved these errors in the revised manuscript in which all the figures are correctly labeled.

6. In Figure 3, the mHtt did not have much more translation inhibition than wtHTT. It requires higher concentration/amount to show slight effects. This cannot explain the profound translation inhibition observed in vivo, as in Fig. 1 and 2. Particularly, mHtt level was actually much lower than wtHtt as shown in Figure 1C.

We agree that the effect of mHtt in vivo is much stronger than in vitro than wtHtt. We posit that in vivo, it may regulate translation with additional regulators and post-translational modifications, which have been indicated in the revised texts (line 264).

7. The dynamic data in Figure 3D seemed not a proper piece of evidence indicating the polyQ interfering with the translation, but rather a slower deterioration of the luciferase signal compared with the BSA control. Tracking luciferase activity by choosing several time points during the 90min IVT reaction could be a better experiment.

As recommended by the reviewer, we carried out time-dependent effects of HTT/mHTT on luciferase synthesis, and mHTT/wtHTT diminishes luciferase activity (Fig. 3D).

8. The data in Figure 5H is not consistent with Figure 2E. the basal level of PS/MS ratio was not changed in 2E. Similar in Figure 3A (actually Figure 6A), the ribosome profile (PS/MS) presented here showed difference between control and HD-het cells, however contradictory to the data shown in Figure 1B, in which the quantification suggests there is no significant difference.

Although some HD-het profiles (n = 8) appear to display enhanced polysome, they were not statistically significant compared to controls (n = 20) (Fig. 1B), as also indicated in response to comment #1

9. Figure 6B, samples from all fractions need to be run on one gel in order to know the relative distribution and abundance of the proteins. The current way of data presentation missed a lot of information.

As per the reviewer's recommendation, we have now loaded the control and HD samples (vehicle and harringtonine treated) in the same gel. The data shows both wtHtt and mHtt are found in the PS fractions. Upon harringtonine, they move to the lower density fractions, indicating that the HTT and mHTT associate with translating ribosomes (Fig. 6A).

10. Figure 6C, why using puromycin instead of harringtonine, which was used in all the other data? In fractions 3-6, the reduction of FMRP and RPL7 was not obvious as claimed, and the signal was way too saturated to know the relative quantity.

As recommended, we have used harringtonine (Fig. 6A). Please see the response to comment #9.

11. In figure 8A, why HD-heter and HD-homo has opposite changes? According to all other data in the manuscript, HD-heter and HD-homo have similar trend of changes, with higher magnitude in HD-homo. It is hard to understand the rationale of opposite trend of relative positions of ribosomes on mRNA in these two cells. And more importantly, the patients are actually HD-heter. How to understand the relevance of this data to patients? The HD-heter data in Figure 8A would not support the proposed model. Better to perform polysome run-off mRNA-seq in patient cells as well.

The reviewer asks a very intriguing question: why in the HD-homo, the ribosomes appear to be on the 5' end of the transcript, while in HD-het, they are towards the 3' end. Although underlying mechanisms for this phenomenon are currently unknown, we speculate that both wtHtt and mHtt in the same cell may impact ribosome translocation differently from wtHtt alone or mHtt alone expressing cells (discussion line 430). For example, when both wtHtt and mHtt are present, they may compete with the ribosomal resources and altered stoichiometry during elongation. During such competition, mRNA transcripts in HD-het may experience differential ribosome occupancy, but these mechanisms remain to be tested in future experiments. For example, isolation of HTT and mHTT from the polysomes in control, HD-het and HD-homo, and identifying interacting partners may further provide insights.

Although the majority of HD patients are heterozygous, there are homozygous patients found. The onset of symptoms is similar in the homozygous and heterozygotes patients with the same CAG repeat lengths. Still, homozygous cases experience severe clinical courses, such as rapid striatal atrophy, the decline in motor, cognitive, and behavioral symptoms (Maglione et al., 2006). Therefore, we used both HD-het and HD-homo cells in the Ribo-Seq study. Our future goal is to prepare Ribo-Seq and polysome run-off mRNA-seq in hESC patient-derived striatal neurons and mouse striatum further to understand any striatal-specific ribosome stalling mechanisms in HD.

12. The author showed the enrichment of the single paused codon in the polysome run-off mRNA-seq data (Figure 8D). The current manuscript will benefit more by validating and discussing more of the hits from the two sets of sequencing data, such as a

detailed look into if any specific pathway or motif are enriched in those stalled mRNAs targets. The data is not accessible from the link the author provided (Line 318).

We have validated five targets (Fig. 9G) and provided pathway details in the supplementary data (Fig. S10). We included step-wise information to access the UCSC browser link in the revised manuscript material and method section.

Minor points

1. There is no figure legend for Figure 3D.

We have included the legend now Fig. 6B.

2. Figure 5G, the ribosome profiling lines need to use more distinct colors.

We have applied distinct colors recommended by the reviewer and is now Fig. 2 D, G, J.

3. In Line 246-247, “As a positive control, we detected Caprin1, a previously 247 known interactor of mHtt”. Need to cite Figure 6F.

We have cited Fig 6D in the revised manuscript.

4. Figure 4C and Figure 8D the two volcano plots are not very clear. The genes and the corresponding dots need to be more clearly labeled.

We have now clearly labelled the genes in Fig. 4B and Fig. 8D in the revised manuscript.

5. The total Fmr1 RNA level should be measured in Figure 4E.

We found Fmrp protein levels are upregulated in HD (Fig. 4D, E). We added Fmr1 mRNA information from (RNA-seq) of HD-homo cells, which showed slightly increased transcripts levels than control (Fig. 4D). In contrast, qPCR analysis for FMR1 in HD patient tissue resulted in no significant alteration of FMR1 RNA in human HD striatum (Fig. 4E). Thus, we concluded Fmrp protein levels are upregulated in HD.

6. Several typos eg. figure 5E (actually Figure 4E) “wtHtt” instead of “mtHtt” and Figure 8D “targets”.

7. Line 124, double “compared to”

8. Line 410, “interfere” instead of “interferes”

We fixed typos referred to in comments #6-8.

References:

- Arguello, R.J., M. Reverendo, A. Mendes, V. Camosseto, A.G. Torres, L. Ribas de Pouplana, S.A. van de Pavert, E. Gatti, and P. Pierre. 2018. SunRISE - measuring translation elongation at single-cell resolution by means of flow cytometry. *J Cell Sci.* 131.
- Azzam, M.E., and I.D. Algranati. 1973. Mechanism of puromycin action: fate of ribosomes after release of nascent protein chains from polysomes. *Proc Natl Acad Sci U S A.* 70:3866-3869.
- Cisbani, G.C.F. 2012. An in vitro perspective on the molecular mechanisms underlying mutant huntingtin protein toxicity. *Cell Death and Disease* 3:e382.
- Conn, C.S., and S.B. Qian. 2013. Nutrient signaling in protein homeostasis: an increase in quantity at the expense of quality. *Sci Signal.* 6:ra24.
- Dana, A., and T. Tuller. 2012. Determinants of translation elongation speed and ribosomal profiling biases in mouse embryonic stem cells. *PLoS Comput Biol.* 8:e1002755.
- Darnell, J.C., S.J. Van Driesche, C. Zhang, K.Y. Hung, A. Mele, C.E. Fraser, E.F. Stone, C. Chen, J.J. Fak, S.W. Chi, D.D. Licatalosi, J.D. Richter, and R.B. Darnell. 2011. FMRP stalls ribosomal translocation on mRNAs linked to synaptic function and autism. *Cell.* 146:247-261.
- Eastman, G., P. Smircich, and J.R. Sotelo-Silveira. 2018. Following Ribosome Footprints to Understand Translation at a Genome Wide Level. *Comput Struct Biotechnol J.* 16:167-176.
- Ehrnhoefer, D.E., L. Sutton, and M.R. Hayden. 2011. Small changes, big impact: posttranslational modifications and function of huntingtin in Huntington disease. *Neuroscientist.* 17:475-492.
- Fresno, M., A. Jimenez, and D. Vazquez. 1977. Inhibition of translation in eukaryotic systems by harringtonine. *Eur J Biochem.* 72:323-330.
- Ingolia, N.T., L.F. Lareau, and J.S. Weissman. 2011. Ribosome profiling of mouse embryonic stem cells reveals the complexity and dynamics of mammalian proteomes. *Cell.* 147:789-802.
- Lee, S., B. Liu, S. Lee, S.X. Huang, B. Shen, and S.B. Qian. 2012. Global mapping of translation initiation sites in mammalian cells at single-nucleotide resolution. *Proc Natl Acad Sci U S A.* 109:E2424-2432.
- Maglione, V., M. Cannella, R. Gradini, G. Cislighi, and F. Squitieri. 2006. Huntingtin fragmentation and increased caspase 3, 8 and 9 activities in lymphoblasts with heterozygous and homozygous Huntington's disease mutation. *Mech Ageing Dev.* 127:213-216.
- Shcherbakov, D., Y. Teo, H. Boukari, A. Cortes-Sanchon, M. Mantovani, I. Osinnii, J. Moore, R. Juskeviciene, M. Brilkova, S. Duscha, H.S. Kumar, E. Laczko, H. Rehrauer, E. Westhof, R. Akbergenov, and E.C. Bottger. 2019. Ribosomal mistranslation leads to silencing of the unfolded protein response and increased mitochondrial biogenesis. *Commun Biol.* 2:381.
- Sivan, G., N. Kedersha, and O. Elroy-Stein. 2007. Ribosomal slowdown mediates translational arrest during cellular division. *Mol Cell Biol.* 27:6639-6646.
- Subramaniam, S., K.M. Sixt, R. Barrow, and S.H. Snyder. 2009. Rhes, a striatal specific protein, mediates mutant-huntingtin cytotoxicity. *Science.* 324:1327-1330.

REVIEWERS' COMMENTS

Reviewer #1 (Remarks to the Author):

Review for the manuscript NCOMMS-20-05400A by Dr. Subramaniam and co-authors entitled "Mutant huntingtin stalls ribosomes and represses protein synthesis independent of Fragile X mental retardation protein"

The authors have addressed most of my previous concerns. I support the publication of manuscript in Nature Communications.

Reviewer #2 (Remarks to the Author):

Review of the Nature communication # NCOMMS-20-05400A

Mutant Huntingtin Stalls Ribosomes and Represses Protein Synthesis in a Cellular Model of Huntington disease

This paper have used a number of advanced technologies to study the Huntington disease cell models. The author's reported a new function for the huntingtin protein and a new mechanism for mutant HTT in the HD pathogenesis.

They have been very responsive to the reviewers and performed new experiments to provide with new results. I am satisfied with their answering to my questions. I have no further comments.

Reviewer #3 (Remarks to the Author):

In the revised version of the manuscript, the authors addressed most concerns and significantly improved the manuscript. Two points:

1. In Figure 6A, there is no data on FMRP, though it was mentioned in results. The data should be included in the figure. Also it said "Rpl6" in the result, but "Rpl7" in the figure. Furthermore, the shift of Rpl7 and Rpl35A is very subtle, not comparable to Htt. This is hard to explain. And there is an extra band in lane 5, which was not explained.
2. In figure 9, it is nice to include examples of candidate genes. But 9A-F is confusing. The Y-axis have very different scales in different cells for each gene. Does it mean the total RNA expression levels as well as the total ribosome occupancy on RNAs are different in control and HD cells? Then the protein level changes shown in Figure 9G might not only be caused by the ribosome stalling (distribution changes) but primarily by RNA expression and/or translation initiation changes. Same scales for the same gene should be used in order to compare in different cells. How the data is normalized need to be included in the legend. The total RNA levels also need to be taken into consideration and included in the figure.

Reviewers #1 and #2 had no additional suggestions. Reviewer #3 additional minor points, which are addressed as indicated below.

Reviewer #3 (Remarks to the Author):

In the revised version of the manuscript, the authors addressed most concerns and significantly improved the manuscript. Two points:

1. In Figure 6A, there is no data on FMRP, though it was mentioned in results. The data should be included in the figure. Also it said "Rpl6" in the result, but "Rpl7" in the figure. Furthermore, the shift of Rpl7 and Rpl35A is very subtle, not comparable to Htt. This is hard to explain. And there is an extra band in lane 5, which was not explained.

Response: We corrected the text in the result to Rp7 and added FMRP, as recommended by reviewer # 3. We reasoned in the result why Rp7 and Rpl35A shift subtle "because Htt is less abundant on polysomes than ribosomal proteins (line 255)."

The extra band in lane 5 is due to "due to the overloading of monosome-accumulated proteins from the collected fraction (lane 252)."

2. In figure 9, it is nice to include examples of candidate genes. But 9A-F is confusing. The Y-axis have very different scales in different cells for each gene. Does it mean the total RNA expression levels as well as the total ribosome occupancy on RNAs are different in control and HD cells? Then the protein level changes shown in Figure 9G might not only be caused by the ribosome stalling (distribution changes) but primarily by RNA expression and/or translation initiation changes. Same scales for the same gene should be used in order to compare in different cells. How the data is normalized need to be included in the legend. The total RNA levels also need to be taken into consideration and included in the figure.

Response: Reviewer raised an important point. In the revised manuscript, we provided RPF, RNA and RPF/RNA ratio of all the candidate mRNAs (lane 368-370, Figure 9).